



# Strong warming of subarctic forest soil deteriorated soil structure via carbon loss – Indications from organic matter fractionation

Christopher Poeplau[1], Páll Sigurðsson [2], Bjarni D Sigurðsson [2]

[1]Thünen Institute of Climate-Smart Agriculture, Bundesallee 68, 38116 Braunschweig, Germany

[2]Agricultural University of Iceland, Hvanneyri IS-311, Borgarnes, Iceland

*Correspondence to*: Christopher.Poeplau@thuenen.de

Keywords: soil warming, macroaggregates, temperature sensitivity, bulk density, soil organic matter

**Abstract.** Net loss of soil organic carbon (SOC) from terrestrial ecosystems is a likely consequence of global
warming and this may affect key soil functions. Strongest changes in temperature are expected to occur at high
northern latitudes, with boreal forest and tundra as prevailing land-cover types. However, specific ecosystem
responses to warming are understudied. We used a natural geothermal soil warming gradient in an Icelandic
spruce forest (0-17.5 °C warming intensity) to assess changes in SOC content in 0-10 cm (topsoil) and 20-30 cm
(subsoil) after 10 years of soil warming. Five different SOC fractions were isolated and the amount of stable
aggregates (63-2000 µm) was assessed to link SOC to soil structure changes. Results were compared to an
adjacent, previously investigated warmed grassland. Soil warming had depleted SOC in the forest soil by -2.7 g
kg$^{-1}$ °C$^{-1}$ (-3.6 % °C$^{-1}$) in the topsoil and -1.6 g kg$^{-1}$ °C$^{-1}$ (-4.5 % °C$^{-1}$) in the subsoil. Distribution of SOC in
different fractions was significantly altered, with particulate organic matter and SOC in sand and stable
aggregates being relatively depleted and SOC attached to silt and clay being relatively enriched in warmed soils.
The major reason for this shift was aggregate break-down: topsoil aggregate mass proportion was reduced from
60.7±2.2 % in the unwarmed reference to 28.9±4.6 % in the most warmed soil. Across both depths, loss of one
unit SOC caused a depletion of 4.5 units aggregated soil, which strongly affected bulk density (R²=0.91 when
correlated to SOC and R²=0.51 when correlated to soil mass in stable aggregates). The proportion of water
extractable carbon increased with decreasing aggregation, indicating an indirect SOC protective effect of
aggregates >63 µm. Topsoil changes in total SOC and fraction distribution were more pronounced in the forest
than in the adjacent warmed grassland soils, due to higher and more labile initial SOC. However, no ecosystem
effect was observed in the response of subsoil SOC and fraction distribution. Whole profile differences across
ecosystems might thus be small. Changes in soil structure upon warming should be studied more deeply and
taken into consideration when interpreting or modelling biotic responses to warming.

**1 Introduction**

Global warming is inexorably progressing, with largest expected changes to occur in high northern latitudes
(Diffenbaugh and Giorgi, 2012). The IPCC worst case scenario (RCP 8.5), predicts a temperature increase of up
to 11°C in areas North of 60° latitude until the end of this century (IPCC, 2013). This will lead to strong



responses of ecosystems, one of which being increased microbial activity and thus oxidation of carbon (Melillo et al., 2002). Predicted alterations in soil organic carbon (SOC), as the largest terrestrial carbon (C) pool (Scharlemann et al., 2014), are inducing a positive climate- carbon cycle feedback loop. The highest SOC stocks, partly associated with thawing permafrost, are located in high northern ecosystems (Tarnocai et al., 2009). This

spatial coherence of the strongest warming and the highest SOC stocks is expected to turn the vast land masses in high northern latitudes into a major C source. Simple extrapolations of short-term soil warming experiments predicted a global SOC loss of up to 203±161 Pg C with 1 °C warming until 2050 (Crowther et al., 2016), which equals one fourth of the current atmospheric C pool. More conservative estimates of the same authors still predicted losses of 55±50 Pg C. This range in possible SOC changes, as well as the large standard errors

associated to each of the estimates points towards the high uncertainty of potential changes in carbon fluxes from terrestrial ecosystems to the atmosphere (van Gestel et al., 2018).

One of the major uncertainties in predicting SOC responses to warming is due to an incomplete mechanistic understanding of the temperature sensitivity of different functional SOC pools. For example, owing to different methodological approaches and partly also misinterpretations (Conant et al., 2011), slow-cycling SOC is found

to be more (Lefevre et al., 2014) or equally (Fang et al., 2005) sensitive to warming than fast-cycling SOC. In consequence, SOC models frequently use the same temperature sensitivity for all SOC functional pools. However, it has been suggested lately that the implementation of carbon turnover and stabilization in many models is outdated (Bradford et al., 2016) and that more wholistic experimental knowledge on warming-induced mechanisms related to carbon turnover in soils is necessary  (Conant et al., 2011). Isolated quantifications of

$CO_2$ fluxes, bulk SOC or even SOC fractions might thus not yield enough insights to understand and predict SOC dynamics under global warming. Furthermore, individual soil warming experiments are mostly restricted to one ecosystem type and differ strongly in methodology, i.e. type and degree of warming. Comparisons across ecosystems are thus hampered (Crowther et al., 2016), but might be critically important to i) foster the understanding of underlying processes driving SOC responses to warming and ii) inform land-surface models to

increase their accuracy.

Apart from its significant role in the global carbon cycle, soil organic matter has numerous functions related to soil fertility and soil health: It is an important food source for soil biota (Barrios, 2007), contains and binds major plant nutrients and trace elements, has a large water storage capacity and is directly linked to soil structure, i.e. the three-dimensional arrangement of soil particles and pore space (Larsbo et al., 2016). Soil

structure drives water and gaseous fluxes through the soil matrix, root growth and nutrient uptake as well as soils susceptibility of soils to compaction and erosion (Johnston et al., 2009;Chepil, 1951;Horn et al., 1994). In addition to the enrichment of atmospheric $CO_2$, soil carbon loss upon warming might thus also deteriorate soil quality, with potential consequences for net primary production. To date, such effects, and involved mechanisms, have been little-noticed, which might be related to the fact that most warming experiments were only run for a

relatively short period of time and with moderate warming treatments (Rustad, 2001;Conant et al., 2011).

In essence, long-term multi-ecosystem warming studies with strong soil warming gradients that might even exceed realistic temperature changes are ideal for advancing our understanding of carbon cycling and related changes in soil functions under global change (Kreyling et al., 2014). Such an experiment has been established in southern Iceland, where an earthquake in 2008 shifted geothermal channels within the bedrock, resulting in



strong gradients in soil warming (up to ~80°C) in previously unwarmed grassland and forest soils. A growing community of scientists is investigating warming effects in permanent monitoring plots on virtually all ecosystem aspects since 2013 (www.forhot.is). In a previous study, Poeplau et al. (2017) quantified the effect of soil warming on bulk SOC and five different SOC fractions with distinct turnover rates in the unmanaged grassland soil. The authors found a strong decline of soil mass and C in the stable aggregate fraction, indicating that either i) warming-induced SOC depletion led to a destabilization of aggregates or ii) warming-induced aggregate break-down led to a destabilization of SOC.

In this study, we isolated the identical SOC fractions from an equally warmed adjacent forest soil to i) advance our understanding of the temperature response of different SOC fractions representing kinetic pools, ii) assess the role of the ecosystem type in the temperature response of SOC and iii) investigate potential links between SOC loss and soil structure changes.

## 2 Materials and methods

### 2.1 Study site

In May 2008, a major earthquake in southern Iceland affected geothermal channels close to its epicenter (Halldórsson et al., 2009). Thereby, a geothermal system in Reykir, close to the village of Hveragerði (64.008°N, 21.178°W) was moved to a previously unwarmed area, which is now constantly warmed in strong temperature gradients of up to ~80°C (O'Gorman et al., 2014). This recently warmed area is coverd by a Sitka spruce forest (*Picea sitchensis* (Bong.) Carr.) that was planted in 1966 and an adjacent unmanaged treeless grasslands dominated by common bent (*Agrostis capillaris*, L.). Those two ecosystems are located on a southwest sloping hill-slope (83-163 m a.s.l.). Mean annual temperature and precipitation between 2003 and 2015, as measured at the closest weather station, were 5.2 °C and 1457 mm respectively (Sigurdsson et al., 2016). According to the world reference base, the soil is characterized as a Silandic Andosol with a silt loam texture (clay:silt:sand:ratio of 8:61:31 in the forest and 6:53:41 in the grassland) (Sigurdsson et al., 2016). Soil pH is slightly acidic (5.3 in the forest and 5.7 in the grassland) and average SOC contents in 0-10 cm soil depth in the unwarmed soils are 75 g kg$^{-1}$ in the forest (present study) and 54 g kg$^{-1}$ in the grassland (Poeplau et al., 2017). Between autumn 2012 and spring 2014, a total of 30 permanent plots were installed in each ecosystem, comprising six different degrees of warming along five different transects. In 2014, the permanently monitored average soil temperature changes due to geothermal warming were 0, 1.0, 1.9, 2.7, 5.8 and 17.5°C in the forest and 0, 0.5, 2.1, 3.9, 10.5 and 17.3°C in the grassland (Sigurdsson et al., 2016).

### 2.2 Soil sampling, fractionation and analysis

In late April 2018, i.e. almost exactly 10 years after the warming was initiated, mineral soils of all permanent plots in the forest were sampled. Before sampling, the litter layer was carefully removed. Sampling was done with a thin auger (3 cm diameter) to a depth of 30 cm in direct proximity of the plot. For each plot, three individual soil cores were taken, split into 0-10, 10-20 and 20-30 cm depth increments and pooled per depth. For this study, only 0-10 cm and 20-30 cm depth increments were used, which will hereafter be referred to as topsoil and subsoil. After sampling, soils were oven dried at 40°C and sieved to <2 mm.



Fractionation of SOC was performed as initially described by Zimmermann et al. (2007) and refined by Poeplau et al. (2013). A scheme can be found at https://www.somfractionation.org/combined-meth/part-dens-oxid-zimmermann/. The procedure involves chemical (oxidation) and physical (size and density separation) fractionation steps, based on current understanding of prevailing SOC stabilization mechanisms in soils. In a

recent comprehensive method comparison, this method was among the most efficient to isolate SOC fractions with varying turnover rates (Poeplau et al., 2018). Distinct responses to warming were thus expected. In brief, 20 g of sieved soil were suspended in 150 ml deionised water and subjected to a light ultrasonic treatment of 21 J ml$^{-1}$ at 30 W to disperse the most instable aggregates and associations. Subsequently, soil was wet sieved with a fixed amount of water over 63 µm to separate silt and clay-sized particles from sand-sized particles. Several

pretests with the most extreme warming treatments and the unwarmed reference revealed that 1400 ml of deionized water was sufficient for a complete separation of coarse (>63 µm) and fine fraction (<63 µm) particles, as indicated by clear rinsing water. The fine fraction containing silt and clay-sized particles (SC) was centrifuged for 15 minutes at 1000 g and an aliquot of the supernatant was filtered over 0.45 µm to derive the dissolved organic carbon fraction (DOC). Fine and coarse fractions were oven-dried at 40°C and weighed.

Sodium polytungstate (SPT) with a density of 1.8 was used to separate the coarse light fraction, i.e. particulate organic matter (POM), from the coarse heavy fraction, i.e. the sand and stable aggregates fraction (SA). To do that, about 40 ml SPT was added to the coarse fraction in a centrifuge tube and stirred gently. Stirred samples were left standing for several hours in room temperature so that particles could float or sink and subsequently centrifuged for 15 minutes at 1000 g for complete separation of light and heavy fractions. The supernatant was

decanted into a sieve bag of 50 µm mesh size. The density fractionation procedure was repeated once to ensure complete separation of light and heavy fractions. After the second SPT treatment, the remaining heavy fraction was transferred to a sieve bag of 50 µm mesh size and both heavy and light fractions were washed thoroughly to remove all SPT, dried at 40°C and weighed. Based on this procedure, we use the term aggregates in the following for the 63-2000 µm aggregate size fraction, which comprises larger microaggregates as well as

macroaggregates (Totsche et al., 2018). Finally, the SC fraction was subjected to sodium hypochlorite (NaOCl) oxidation, which is done to mimic strong enzymatic decay and isolate an oxidation-resistant SOC fraction (rSOC). To do so, NaOCl with 6 % Cl was first adjusted to pH 8 using concentrated HCl. A 1 g aliquot of the SC fraction was then mixed with 40 ml NaOCl. After 17 hours reaction time, samples were centrifuged, decanted and washed once with deionised water. The whole procedure was repeated twice to ensure complete oxidation of

NaOCl-oxidizable SOC (SC-rSOC). Thereafter, soil was dried at 40°C and weighed to determine the mass loss caused by oxidation. All solid fractions and the bulk soil were ball-milled and measured for C and N contents via dry combustion (LECO-TruMac, St Joseph, MI, USA). The DOC fraction was measured using a liquid analyser (DIMATOC, Dimatec, Essen, Germany). Average mass recovery was 97%, average C recovery was 99%. In the following, two different measures of SOC in the isolated fractions will be used, depending on the context: i)

SOC concentration, which indicates the amount of SOC in each fraction per fraction mass [g C kg fraction$^{-1}$], and ii) SOC content, which indicates the amount of SOC in each fraction per bulk soil mass [g C kg soil$^{-1}$].

To determine the total amount of soil in stable aggregates, i.e. to separate the SA fraction into sand and stable aggregates, another 4 g of each bulk soil sample was used posterior. Instead of the soft ultrasonic treatment of 21 J ml$^{-1}$, we applied 500 J ml$^{-1}$ at a high amplitude (70%) to completely disperse all aggregates (Schmidt et al.,

1999). After subsequent wet sieving, the mass proportion of the coarse fraction (>63 µm) containing POM and



pure sand grains was determined and subtracted from the earlier coarse fraction to determine the mass proportion of stable aggregates.

To evaluate the effect of bulk SOC and SOC fractions on soil structure, we determined the poured bulk density in the bulk soil as well as the coarse (SA+POM) and fine (SC) fractions of each sample. Poured bulk density is

also known as aerated bulk density and is a measure of structural strength of loose material (Abdullah and Geldart, 1999). This was done by pouring the material of known weight into a scaled cylindric flask to measure the volume of the sample. Poured bulk density of each individual sample ($\rho_i$, g cm$^{-3}$) was then calculated as:

$$\rho_i = \frac{mass_i}{volume_i}$$
(Eq. 1),

where $Mass_i$ is the total soil mass of the individual fraction [g] and $Volume_i$ is the volume of the individual

fraction [cm$^{-3}$]. We assumed that a higher poured bulk density would indicate less structure and hypothesized that $\rho_i$ would be particularly correlated to SOC content in the SA fraction.

Soil sampling of the adjacent grassland SOC (data from previous study) was done in 2014, six years after the the warming was initiated, and involved the same analyses as was done on the forest soil (Poeplau et al., 2017).

### 2.3 Statistics

The balanced design of the experiment, i.e. six warming intensities, five transects (replicates) and two different sampling depths, allowed the use of analysis of variance (ANOVA) to test differences between warming intensities in bulk SOC and SOC fractions for significance. Also, non-parametric analysis of similarity (ANOSIM) as implemented in the R package vegan (Oksanen et al., 2019) was used to test if warming significantly altered SOC composition, i.e. its distribution in different fractions. Finally, analysis of covariance

was used to assess, whether forest SOC (data from this study) and grassland SOC (data from previous study) would differ in their response to soil warming. This was done using linear regression models including ecosystem, warming intensity and their interaction. Linear regressions were also used to derive absolute and relative changes in SOC content per °C. Whenever necessary, data was log-transformed to approximate normal distribution, which was visually assessed using histograms. Significance was assessed at a level of $p<0.05$. All

statistical tests and plots were done in R (R Development Core Team, 2010). For plots, the package ggplot2 was used (Wickham, 2016).

## 3 Results

### 3.1 Warming induced changes in forest soil organic carbon

After ten years of soil warming, bulk SOC content in the forest soil had dropped severely in all investigated

warming treatments. Warming induced SOC losses increased linearly with degree of warming (Fig.1A, 1B, Tab. 1) in both depth increments. Absolute losses in the topsoil (-2.7 g kg$^{-1}$ °C$^{-1}$, Tab. 1) were more pronounced than absolute losses in the subsoil (-1.6 g kg$^{-1}$ °C$^{-1}$). In the topsoil, SOC dropped from 75.1 g kg$^{-1}$ in the unwarmed soil to 26.5 g kg$^{-1}$ in the most warmed soil; in the subsoil it dropped from 36.2 g kg$^{-1}$ to 4.0 g kg$^{-1}$. Relative losses were thus even more pronounced in the subsoil (-4.5 % SOC °C$^{-1}$) as compared to the topsoil (-3.6 % SOC °C$^{-1}$).





Despite these strong linear trends, SOC were only significantly different from the unwarmed reference at a warming intensity of 5.8°C and 17.5°C (topsoil) as well as 17.5°C (subsoil) (Tab. 1). The same was true for SOC concentrations in SA and POM, while for SC and rSOC only a warming intensity of 17.5 °C was enough to significantly decrease SOC concentration in both depths after 10 years. For DOC, significant changes with warming were only observed in the subsoil. In the topsoil, relative changes in SOC content were in the order POM > SA > bulk soil > DOC > SC > rSOC, which is in agreement with the concept of the fractionation method, i.e. a stronger decline in the most labile fractions and a slower decline in the more stable fractions. However, this was not the case for the subsoil, in which the order of relative SOC changes almost reversed to rSOC > SC > POM > bulk soil > SA>DOC (Tab. 1). The strong changes in rSOC and SC were however mainly driven by the 17.5°C warming intensity.

Depletion of SOC led to a changed relative distribution of the isolated fractions with warming level (Fig. 1C, 1D). The ANOSIM revealed that a warming intensities of 5.8 and 17.5°C were necessary to significantly change SOC distribution (Tab. 2). In the subsoil, fraction distribution was only significantly different from the unwarmed reference at a warming intensity of 5.8°C. The unwarmed reference soil was strongly dominated by SOC in the POM and SA fractions (together ~90 %), which were strongly depleted with warming (Fig. 1). This led to a relative increase of SOC stored in the fine fractions (SC-rSOC and rSOC). In the topsoil, even an absolute increase of SOC in these fractions was observed upon warming (Fig. 1A), which strongly indicated a redistribution of fraction masses. Indeed, the soil mass of the SA fraction decreased with warming, while the mass of the SC fraction increased (Fig. 2). This was true for both investigated soil depths, with the mass distribution of the subsoil at 17.5°C warming intensity being an exception. As expected, the second ultrasonic step revealed that within the SA fraction, only the aggregates depleted, while the proportion of sand sized mineral particles remained stable across warming levels (Fig. 2). Thereby, aggregate mass proportion in the topsoil decreased from 60.7±2.2 % in the unwarmed reference to 28.9±4.6 % in the 17.5°C warmed soil. In the subsoil, it decreased from 43.7±3.8 % in the unwarmed reference to 17.7±2.9 % in the 5.8°C warmed soil, while at a warming intensity of 17.5°C the mass proportion of aggregates amounted to 32.9±4.9 %. The average sand content of 28 % determined after the second ultrasonic treatment (Fig. 2) was well in line with the 31 % sand content of the texture analysis.

Within the fine fraction, the relative proportion of rSOC was expected to increase with warming, due to its proposed higher biogeochemical stability as compared to the NaOCl-oxidised part of the SC fraction. This was however not the case: Across all warming intensities and both soil depths, we found a significant linear correlation between rSOC and total SOC in the SC fraction (y=0.319x, R²=0.92). Thus, the NaOCl treatment did constantly oxidize two thirds of the SC fraction across all warming intensities, indicating that no relative accumulation of rSOC within the silt and clay sized soil fraction occured.

Interestingly, the proportion of SOC that was water soluble (DOC), tended to increase with warming in both investigated depth increments (Fig. 1C and D), which was not significant. However, for the topsoil, we detected a significantly negative relationship of the propotion of SOC in SA and the proportion of SOC in DOC (Fig. 3), which might point towards the SOC stabilizing function of aggregates.




### 3.2 Forest vs. grassland soil carbon responses to warming

The observed changes in bulk and fraction SOC in the forest soil were generally comparable to those in the adjacent grassland soils (Figure 4 and 5). Especially in the subsoil, the interaction effect of ecosystem and warming on SOC was not significant for four out of five fractions and the bulk soil, indicating the same SOC response to warming in both ecosystem types (Fig. 5, Tab. 3). Also, SOC contents in the subsoil were similar for both ecosystems. This might partly be related to the fact that the forest was planted on an unmanaged grassland and that the forest subsoil SOC was still grassland-derived to a high extend. However, for the topsoil we found significant interactive effects of ecosystem and warming for four out of five fractions and the bulk soil (Tab. 3). The forest soil, which had a considerably higher bulk SOC content in the unwarmed reference than the grassland, showed a stronger response to warming. The predominant SOC fraction in the forest topsoil was the SA fraction, which responded strongest to warming (Fig. 1). This was generally observed in both ecosystems. However, the stronger redistribution of soil mass across fractions in the forest soil as compared to the grassland soil led to very distinct responses of SC-rSOC and rSOC, with stronger warming induced increases of these fractions in the forest soil (Fig. 4). Also the POM fraction of the forest soil responded more negatively to warming than that in the grassland soil. Only for the warming response of DOC, we did not detect any differences between ecosystems in the topsoil. Interestingly, despite differences in initial SOC and warming duration, i.e. ten years for the forest and six years for the grassland, SOC in both ecosystems approached an almost equal SOC content in the most extreme warming intensities (Fig. 4).

### 3.3 Structural changes following soil carbon loss

As expected, we found a strong negative correlation of SOC content and poured bulk density (Fig. 6A). A very similar relationship with identical slope was observed for the coarse (>63 µm) soil fraction, comprising SA and POM (Fig. 6B). In contrast to that, we did not detect any correlation of SOC content and poured bulk density in the silt and clay fraction (data not shown). A direct link of poured bulk density and aggregates is given in Fig. 6C. Finally, in agreement with the strong decline of SOC and soil mass in the SA fraction with warming intensity (Fig. 1, 2), we found a strong positive correlation of SOC mass and soil mass in the coarse soil fraction comprising SA and POM (Fig. 6D). The slope of the regression was 4.5, indicating that one unit SOC was causing the aggregation of 4.5 units of soil. The effects of SOC on soil structure were equally observed in topsoil and subsoil. Furthermore, for all structure-related parameters shown in Figure 6, observations of both investigated soil depths scattered approximately around the same regression line. This might indicate that SOC depletion as such, rather than soil warming, induced the break-down of aggregates.

## 4 Discussion

### 4.1 Warming effects on forest soil organic carbon and its fractions

Ten years of forest soil warming caused a strong decline in SOC content. Along the temperature gradient, SOC changes followed a linear response, with -2.7 % and -4.5 % change per °C in topsoil and subsoil, respectively. In the most extreme warming intensity of +17.5°C, SOC was thus depleted by 65 and 89 %. With a warming intensity of 5.8°C, which can be considered a realistic temperature increase in high northern latitudes until the end of the century, the investigated soil lost 29 and 37 % SOC in ten years. This is in line with other studies,





which also reported significant losses of SOC upon warming (Crowther et al., 2016 and papers cited therein). In the investigated experiment, there is no doubt that potential warming-induced changes in net primary productivity (NPP; Sigurdsson et al., 2014) did not offset increased soil microbial activity. Similar or relatively even more pronounced losses of SOC from the subsoil as compared to the topsoil are confirmed by results of a

recent whole profile forest soil warming study, concluding that subsoils will be an important source of $CO_2$ under climate change (Hicks Pries et al., 2017). Higher relative losses of SOC in the subsoil could potentially be driven by warming-induced changes in C input patterns. Indeed especially fine root production and turnover of trees in the boreal zone was previously found to increase with moderate warming (Leppälammi-Kujansuu et al., 2014;Majdi & Öhrvik, 2004), but at the investigated site the amount of fine roots and mycorrhizal production

has been found to decrease at the more extreme warming levels (Parts et al., 2014;Rosenstock et al., 2019). Further, the fine roots are primarily located in the uppermost cm of forest soils (Hansson et al., 2013;Leppälammi-Kujansuu et al., 2013). Losses via SOC mineralization might have thus been buffered to a higher extent by C input in the topsoil as compared to the subsoil. In addition, warming was slightly more intense in the subsoil, since soils are heated from below, even if the vertical gradients within the top 30 cm of

soil were not substantial except in the highest warming level (Sigurdsson et al., 2016).

A major strength of a warming gradient approach is the identification of potential tipping points, which may mark abrupt changes in ecosystem functionality (Kreyling et al., 2014). However, the present study did not reveal such tipping points for SOC, which changed surprisingly linear with increasing temperature in both investigated depth increments. Despite certain methodological drawbacks of the geothermal (or any other

manipulated) soil warming experiment, such as very abrupt initial temperature changes, as well as soil warming from below instead of whole ecosystem warming from above, it can be inferred that climate change is likely to strongly affect SOC stocks of boreal forests. The latter cover an area of approximately 15 mio km² or one third of the global forest area (Bonan, 2008). The analysis of the soil warming gradient also revealed detection limits for warming effects on SOC that is per se very heterogeneous in space and responds slowly to environmental

change (Smith, 2004): Even after ten years of chronic soil warming, changes in topsoil SOC were only significant at a warming intensity of at least 5.8°C, when assessed using the ANOVA approach. The latter, instead of a regression analysis, needs to be used when only one warming treatment is investigated (e.g. Schnecker et al. 2016). If this treatment is relatively mild, e.g. below 4°C, changes might easily be undetectable against the background heterogeneity of SOC. This is an important insight considering the ongoing debate if

SOC is lost upon warming or not (Crowther et al., 2016;van Gestel et al., 2018). The majority of currently available datasets are based on such experiments with relatively short, mild and singular warming treatments (van Gestel et al., 2018).

The fractionation method used in this study isolates SOC pools of different biogeochemical stabilities (Zimmermann et al., 2007). Turnover rates are estimated to range from several years in the POM fraction to

centuries in the oxidation resistant rSOC fraction that is associated to silt and clay particles (von Lützow et al., 2007). Such differences are mainly related to different degrees of physico-chemical stabilization in the soil, such as the interaction with the mineral phase or occlusion into aggregates (von Lützow et al., 2007). Due to differences in composition and bioavailability of these SOC fractions, distinct responses to warming were expected in the order POM > DOC > SA > bulk soil > SC-rSOC > rSOC. Indeed the average relative decrease in

SOC content, which might be the best indicator to describe a fraction's sensitivity to warming, was observed to



follow a similar order in the topsoil: POM > SA > bulk soil > DOC > SC-rSOC > rSOC. This is well in line with the sensitivity of these fractions to land-use change as observed across different land-use changes by Poeplau and Don (2013). The difference in warming response between SC-rSOC (-2.14 % °C$^{-1}$) and rSOC (-2.05 % °C$^{-1}$) was however negligible, which was also reflected in the stable proportion of rSOC in the total SC fraction

throughout the warming gradient. This indicated that NaOCl-oxidation did not yield a meaningful fraction with regard to biogeochemical resistence. This has been observed before and questions the use of chemical oxidation to mimic biological oxidation (Lutfalla et al., 2014;Poeplau et al., 2019;Poeplau et al., 2017). Paradoxically, many authors describe the NaOCl-resistant SOC as substantially older and thus slower cycling as bulk SOC (Helfrich et al., 2007;Mikutta et al., 2005). In any case, the proposed function of rSOC as centenially persistent

or even inert SOC pool (Zimmermann et al., 2007) could repeatedly not be confirmed in this study. In the subsoil, the average relative depletion in rSOC was even strongest across all fractions and the bulk soil. This was however related to the very low carbon content of the highest warming intensity (17.5 °C), driving the slope of the regression. Only when the highest warming intensity was excluded, the sensitivity of fractions followed the observed order in the topsoil, with DOC being an exception: POM > SA > bulk soil > SC-rSOC > rSOC > DOC.

**4.2 Aggregate break-down induced by soil organic carbon losses or vice versa?**

The most significant warming effect on the distribution of SOC in the isolated fractions was the strong decrease of SA. In the unwarmed reference soil, it accounted for the highest proportion of soil mass and SOC. However, with warming, aggregates collapsed, leading to strong mass increases in the fine SC fractions, which even increased in carbon mass upon warming. The second ultrasonic step, which was used to distinguish sand from

aggregates in the SA fraction, provided evidence that the investigated aggregate size fraction (63-2000 μm) was strongly reduced. The same mechanism, yet less pronounced, was observed for the adjacent grassland (Poeplau et al., 2017). Observing SOC depletion and aggregate break-down at the same time rises the question of cause and effect: Aggregates – at least micro-aggregates < 250 μm - are acknowledged to protect organic matter from microbial decomposition (Six et al., 2002). At the same time, organic matter, especially mucilage,

polysaccharides and fungal hyphae acts as aggregate binding agent (Tisdall and Oades, 1982). Answering the question whether warming per se has fostered aggregate break-down through changes in biotic and abiotic environmental conditions might be of critical importance for conceptualizing and modeling warming effects on SOC dynamics. However, results of the present study suggest that the major cause of aggregate break-down was not necessarily warming, but could be well described with loss of SOC: we found a very strong correlation of

SOC mass and total soil mass in the coarse soil fraction (comprising POM and SA) – one 1 g kg$^{-1}$ of SOC was keeping 4.5 g kg$^{-1}$ soil aggregated. Thereby, topsoil and subsoil samples scattered approximately around the same regression line, indicating that especially the amount of young SOC, may it be driven by warming intensity or the location within the soil profile, was a very good predictor for the amount of stable aggregates in the soil. This is well known and thus in accordance with the literature (Franzluebbers, 2002;Oades, 1984;Shepherd et al.,

2002). Another reason to doubt that warming-induced aggregate break-down caused destabilization of SOC is the fact that the SOC protection capacity of macro-aggregtaes is debatable (Six et al., 2004). For example, Bischoff et al. (2017) found higher heterotrophic respiration in uncrushed soil as compared to the same soil with crushed macro-aggregates. To some extent, a positive feedback loop, i.e. SOC depletion causing aggregate break-down which in turn causes mineralization of then accesible C might indeed be possible. The fact that the



proportion of water soluble SOC in the topsoil increased with decreasing aggregation, points in this direction. Desorption of carbon compounds from the mineral phase is likely to be fostered by increased surface area.

### 4.3 Linking losses in soil organic carbon to changes in soil structure

In consequence of SOC loss, total pore space decreased strongly as indicated by poured bulk density used as a
proxy for bulk density in undisturbed samples. The latter was unfortunately not determined in the present study. However, the relationship of SOC and poured bulk density was in the range of established pedotransfer functions (PTF) for field bulk density estimation using SOC content. In a literature review comparing different PTF (De Vos et al., 2005), slopes of the regressions model using SOC content [g kg$^{-1}$] to predict soil bulk density [g cm$^{-3}$] ranged from -0.003 to -0.011, while the slope in the present study was -0.005 for both the bulk soil and the SA
fraction. The negative correlation is due to i) a much lower specific gravity of organic matter as compared to mineral particles, but also due to the effect of organic matter on aggregation (De Vos et al., 2005). The variation in slopes, i.e. effect of SOC on bulk density, is most likely related to the soil's capability to form aggregates. In very sandy soils with a single grain stucture, even high organic matter contents do not lead to considerable formation of aggregates so that the organic matter effect on bulk density is mainly restricted to a gravity effect.
Using a two-pool mixing model of mineral particles with a density of 2.5 g cm$^{-3}$ and soil organic matter with a density of 1, i.e. ignoring the structural effect of organic matter, we found a slope of -0.0026. Accordingly, Callesen et al. (2003) reported a PTF for sandy forest soils with a slope of approximately -0.0028 in the range between 0-80 g SOC kg$^{-1}$ (non-linear function). The slope of -0.005 found in this study might thus indicate that approximately 50 % of the SOC effect on poured bulk density can be assigned to a structural effect. Indeed, we
also found a strong negative correlation of the soil mass stored in aggregates and the poured bulk density. To conclude, the slope of the regression between SOC and bulk density, at least in unmanaged soils, might be a good indicator for the aggregation affinity of a soil. Surely, poured bulk density of disturbed and sieved soil can only express a potential and should be treated as such. On the other hand, factors like position in the soil profile that strongly influence the packing density of the soil are canceled out, enabling a direct comparison of topsoil
and subsoil samples.

Strong systematic gradients in SOC content in the same soil, as have been created by the soil warming in our study, are rare and extremely valuable to improve our understanding on organic matter functions. Larsbo et al. (2016) used a natural SOC gradient to evaluate its effect on pore networks, influencing solute and gaseous transport in the soil. Changes in soil structure as induced by the large SOC loss might also affect other key
ecosystem properties, such as NPP (Oldfield et al., 2019), microbial biomass (Walker et al., 2018) or other soil biota. For example, in the adjacent warmed grassland plots, Holmstrup et al. (2018) detected a warming-induced shift in collembola species abundance towards species with smaller body size. While the authors did not explicitly link that to changes in soil structure, an increase in bulk density with associated decrease in pore space might have fostered that change. Also, a positive correlation of pore volume and microbial and nematode
biomass was found by Hassink et al. (1993). In the present study, aggregation and poured bulk density were assessed on sieved soils, which provided valuable first information on warming-induced changes in basic soil structural parameters. For two major reasons, a follow-up study should investigate soil structure and other physical parameters in undisturbed soil samples: i) the gradient in SOC content is unique and can be used to improve the general understanding of the link between organic matter and soil functions; ii) the warming





responses of many ecosystem aspects are studied along the investigated warming gradients and knowledge on changes in soil physical properties might be central to interpret such responses.

### 4.4 Comparing forest and grassland soil carbon responses to warming

To date, warming experiments have mostly focused on one single type of ecosystem. However, the warming
response could be ecosystem specific (Shaver et al., 2000), which can only be investigated in a paired ecosystem approach. In the present study, we investigated a small stretch of forest located directly adjacent to a similarly warmed grassland. Changes in SOC and SOC fractions in the grassland have been investigated previously (Poeplau et al., 2017). Both ecosystems showed a similarly strong response to warming. The fact that no difference in subsoil SOC dynamics in the bulk soil or any isolated fractions were observed might indicate that
the same mechanisms of SOC depletion were involved. For example, aggregate break-down as well as equal decrease in rSOC and SC-rSOC were also observed in the grassland. However, the initial SOC content and fraction distribution in the topsoil differed across ecosystems, leading to distinct responses to warming: The unwarmed forest had about 50 % more SOC in the topsoil as compared to the grassland, and about 150 % more SOC was stored in the SA fraction. Also the POM fraction was almost doubled in the forest, with proportionally
less SOC stored in more stable fractions. The shift in fraction mass distribution, i.e. aggregtae break-down, was more pronounced in the forest topsoil, leading to the increase in fine fraction SOC with warming, which was not observed in the grassland. Crowther et al. (2016) reported that SOC loss upon warming is a function of initial SOC – the present study confirms that. In fact, to some extent the explanation for that might be the higher proportion of labile SOC in soils with higher SOC stocks (Besnard et al., 1996). It has been reported previously
that forest SOC is more labile than grassland SOC (Poeplau and Don, 2013). The forest was sampled after ten years of warming, the grassland after six years. However, i) subsoils showed an almost identical response to warming and ii) there are indications that at least the grassland reached a new steady state in SOC already after six years of warming (Walker et al., 2018).  Together with the fact that also more labile SOC was present in the forest topsoil, which responded more sensitive to warming than the bulk soil, it seems likely that amount and
fraction distribution of SOC drove the ecosystem specific warming response in the topsoil. The difference in topsoil SOC and fraction distribution was found before and is related to the different sources and qualities of fresh organic matter inputs (Poeplau and Don, 2013;Huang et al., 2011). Especially needle litter is acknowledged to decompose slowly (Prescott et al., 2000). Differences in POM as well as total SOC stocks are observed to level off with increasing soil depth (Davis and Condron, 2002;Poeplau and Don, 2013). This might also be true
for the response to warming, as indicated in the present study.

### 5 Conclusion

Using a strong geothermal warming gradient, we highlighted the critical role of SOC for soil structure. Ten years of soil warming created a steep gradient in SOC contents that is rare and should be used to study the links of organic matter to soil structure and soil functions more deeply. Results of the present study reveal that the effects
of warming on biogeochemical cycles are most likely not restricted to direct effects on biotic processes, but that changes in the soil abiotic environment should be considered. Those are likely to exert a strong indirect influence on any biotic response. Differences in SOC and SOC fractions responses to warming across ecosystems have




been found in the topsoil only, which might however be related to the fact that the forest was planted on unmanaged grassland half a century ago. In the forest, depletion of SOC was more pronounced in the subsoil, which calls for more whole soil profile warming studies.

**Data availiability**

The dataset is stored in the repository of the center for open science and available via DOI 10.17605/OSF.IO/SGUZ2.

**Author contribution**

CP designed the study, carried out parts of the lab work and prepared the manuscript with contributions from all co-authors. PS sampled the soils and BS initiated the entire field experiment.

**Competing Interests**

None.

**Acknowledgements**

We thank Tatjana Saevici for conducting the fractionation work. This work contributes to the Nordic CAR-ES
and the ForHot (www.forhot.is) network projects and the Icelandic Research Fund project 163272-053.

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



Table 1: Average soil organic carbon (SOC) concentrations of all fractions and the bulk soil with standard errors and letters indicating significant differences (p<0.05) across warming intensities [°C] within one soil depth. Absolute and relative changes in SOC content as derived from linear regression models are also displayed for both investigated soil depths. Fractions were dissolved organic carbon (DOC), particulate organic matter (POM),

5    SOC in sand and aggregates (SA), total silt- and clay-sized SOC (SC) and oxidation resistant silt- and clay-sized SOC (rSOC).

| Depth | Warming intensity | Bulk soil | DOC | POM | SA | SC | rSOC |
|---|---|---|---|---|---|---|---|
| | °C | g C kg soil$^{-1}$ | g C kg soil$^{-1}$ | g C kg soil$^{-1}$ | g C kg fraction$^{-1}$ | g C kg fraction$^{-1}$ | g C kg fraction$^{-1}$ |
| Topsoil | 0 | 75.1±5.5a | 0.7±0.1a | 11.8±2.6ab | 6.8±0.5a | 5.1±0.2a | 1.6±0.1a |
| | 1 | 71.5±4.0a | 1.0±0.3a | 21.6±4.4a | 6.3±0.7a | 4.9±0.3a | 1.6±0.1a |
| | 1.9 | 65.9±3.0a | 0.7±0.1a | 12.9±1.8abc | 7.1±1.0a | 5.4±0.2a | 1.9±0.1a |
| | 2.7 | 64.7±1.5ab | 0.6±0.1a | 16.0±3.5ab | 5.8±0.4a | 5.2±0.1a | 1.7±0.1a |
| | 5.8 | 53.1±3.2b | 0.5±0.1a | 6.0±1.5bc | 5.2±0.6a | 5.0±0.3a | 1.8±0.1a |
| | 17.5 | 26.5±1.9c | 0.5±0.1a | 0.4±2.6c | 2.6±0.4b | 3.3±0.2b | 1.1±0.1b |
| Absolute change [g C kg$^{-1}$ fraction °C$^{-1}$] | | -2.71 | -0.02 | -0.84 | -0.25 | -0.11 | -0.03 |
| Relative change [% °C$^{-1}$] | | -3.6 | -2.49 | -7.15 | -3.63 | -2.14 | -2.05 |
| Subsoil | 0 | 36.2±4.3a | 0.3±0.1ab | 3.4±0.8a | 2.9±0.7a | 4.1±0.3a | 1.3±0.1a |
| | 1 | 28.6±4.2a | 0.3±0.1ab | 3.4±0.7a | 1.7±0.4ab | 3.8±0.3a | 1.4±0.2a |
| | 1.9 | 29.4±4.6a | 0.3±0.0ab | 2.0±0.3ab | 1.5±0.4ab | 3.8±0.5a | 1.2±0.2a |
| | 2.7 | 24.2±1.9a | 0.2±0.0ab | 2.1±0.7ab | 1.2±0.1ab | 3.4±0.2a | 1.1±0.1a |
| | 5.8 | 22.6±3.3a | 0.3±0.0ab | 0.8±0.2b | 0.9±0.2b | 3.1±0.4a | 1.1±0.2a |
| | 17.5 | 4.0±0.9b | 0.2±0.0b | 0.3±0.1b | 0.2±0.0c | 0.5±0.2b | 0.2±0.1b |
| Absolute change [g C kg$^{-1}$ fraction °C$^{-1}$] | | -1.63 | -0.01 | -0.16 | -0.11 | -0.2 | -0.07 |
| Relative change [% °C$^{-1}$] | | -4.52 | -2.53 | -4.79 | -3.96 | -4.95 | -5.04 |

10   Table 2: Summary of the analysis of similarity (ANOSIM) testing differences in SOC fraction distribution for all warming intensities tested against the unwarmed reference. P values <0.05 indicate significant differences, while n.s. indicates non significant differences. An R value close to 1 suggest dissimilarity between groups.

| Warming | Topsoil | | Subsoil | |
|---|---|---|---|---|
| [°C] | R | p | R | p |
| 1 | 0.260 | n.s. | 0.040 | n.s. |
| 1.9 | 0.044 | n.s. | 0.168 | n.s. |
| 2.7 | 0.116 | n.s. | 0.380 | 0.044 |
| 5.8 | 0.272 | 0.036 | 0.840 | 0.011 |
| 17.5 | 0.868 | 0.005 | 0.196 | n.s. |



Table 3: Summary of the linear regression models (*p* values) assessing effects of warming, ecosystem (grassland vs. forest) and their interaction on soil organic carbon (SOC) for the bulk soil and all isolated fractions.

| Fraction | Topsoil | | | Subsoil | | |
|---|---|---|---|---|---|---|
| | Warming | Ecosystem | Interaction | Warming | Ecosystem | Interaction |
| Bulk soil | <0.001 | <0.001 | 0.029 | <0.001 | 0.038 | n.s. |
| DOC | 0.016 | n.s. | n.s. | n.s. | 0.001 | n.s. |
| POM | <0.001 | <0.001 | 0.002 | <0.001 | 0.049 | n.s. |
| SA | <0.001 | <0.001 | 0.023 | <0.001 | <0.001 | n.s. |
| SC-rSOC | <0.001 | <0.001 | 0.001 | <0.001 | n.s. | n.s. |
| rSOC | <0.001 | <0.001 | 0.002 | <0.001 | n.s. | 0.042 |

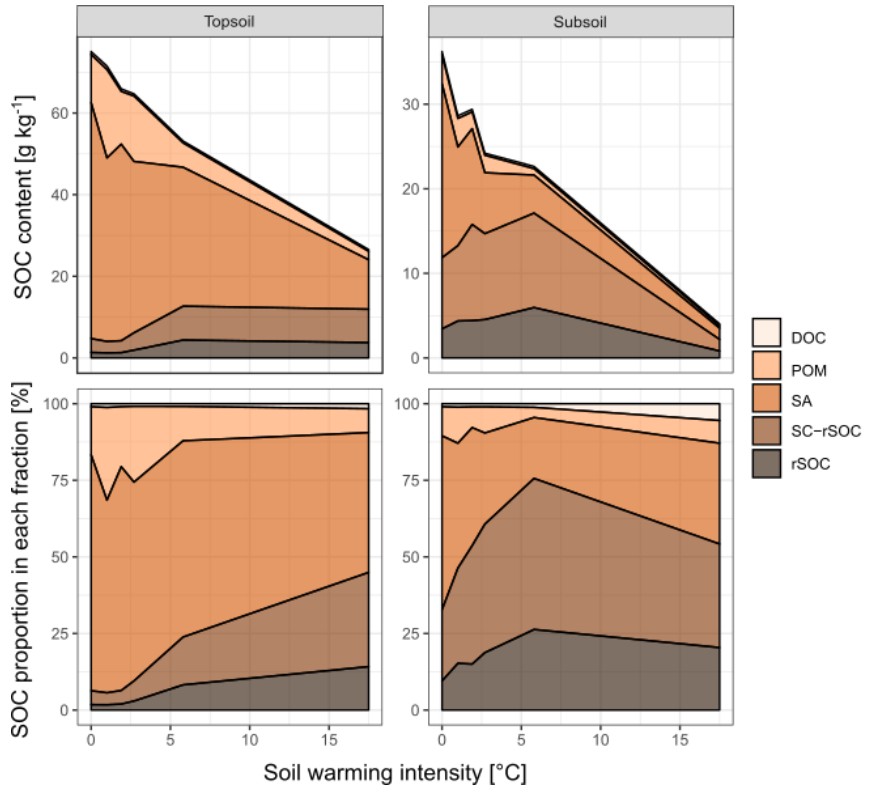

Figure 1: Areal plots of A) soil organic carbon (SOC) content in the topsoil and B) SOC content in the subsoil, C) SOC proportion in each fraction of the topsoil and D) SOC proportion in each fraction of the subsoil as a function of warming intensity. Fractions were dissolved organic carbon (DOC), particulate organic matter (POM), SOC in sand and aggregates (SA), non-oxidation resistant silt- and clay-sized SOC (SC-rSOC) and

10   oxidation resistant silt- and clay-sized SOC (rSOC).



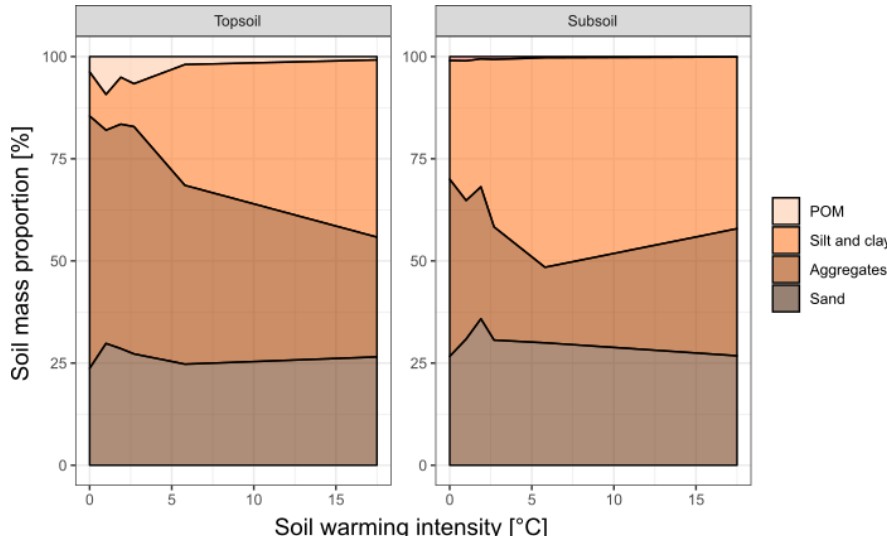

Figure 2: Areal plots of soil mass distribution in the fractions particulate organic matter (POM), sand and stable aggregates (SA) and silt and clay (SC) as a function of warming intensity.

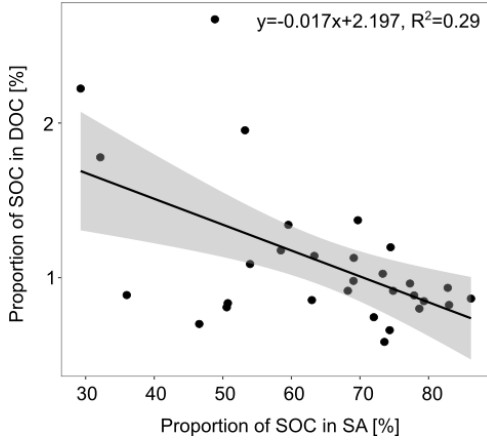

Figure 3: Correlation between the proportion of soil organic carbon (SOC) in the sand and aggregates (SA) and dissolved organic carbon (DOC) fractions.





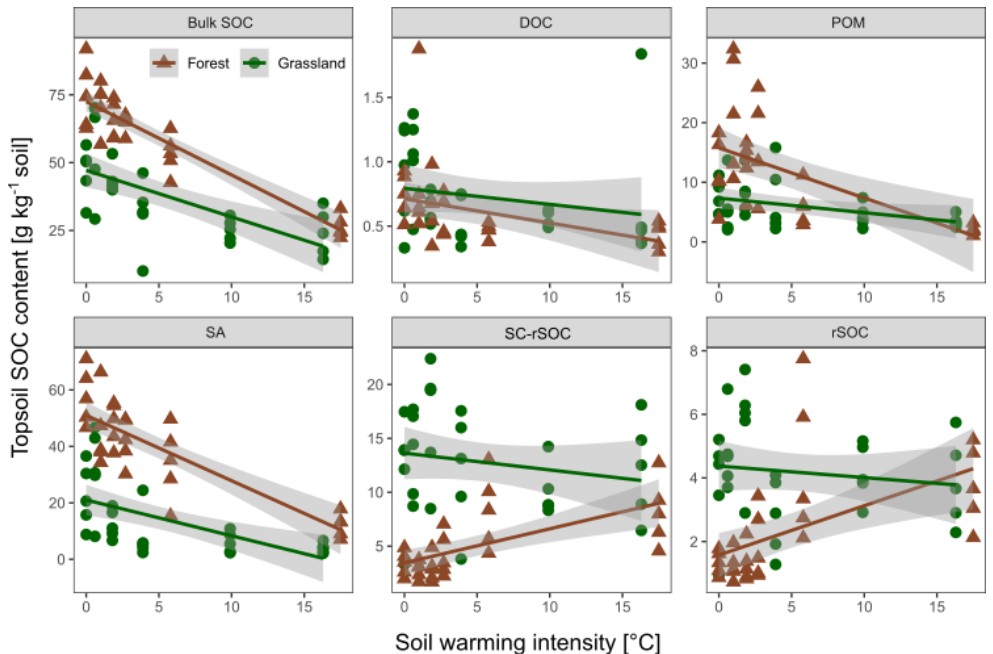

Figure 4: Scatter plots showing soil organic carbon (SOC) mass in bulk soil and fractions of the forest and grassland topsoils (0-10 cm) as a function of warming intensity. Fractions were dissolved organic carbon (DOC), particulate organic matter (POM), SOC in sand and aggregates (SA), non-oxidation resistant silt- and clay-sized SOC (SC-rSOC) and oxidation resistant silt- and clay-sized SOC (rSOC).



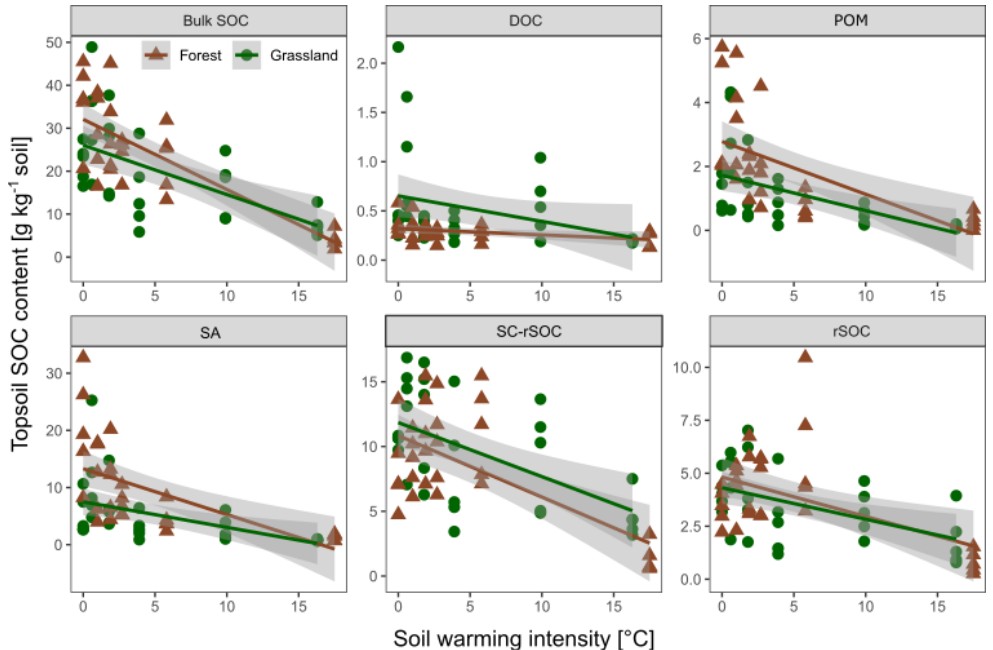

Figure 5: Scatter plots showing soil organic carbon (SOC) content in bulk soil and fractions of the forest and grassland subsoils (20-30 cm) as a function of warming intensity. Fractions were dissolved organic carbon (DOC), particulate organic matter (POM), SOC in sand and aggregates (SA), non-oxidation resistant silt- and clay-sized SOC (SC-rSOC) and oxidation resistant silt- and clay-sized SOC (rSOC).

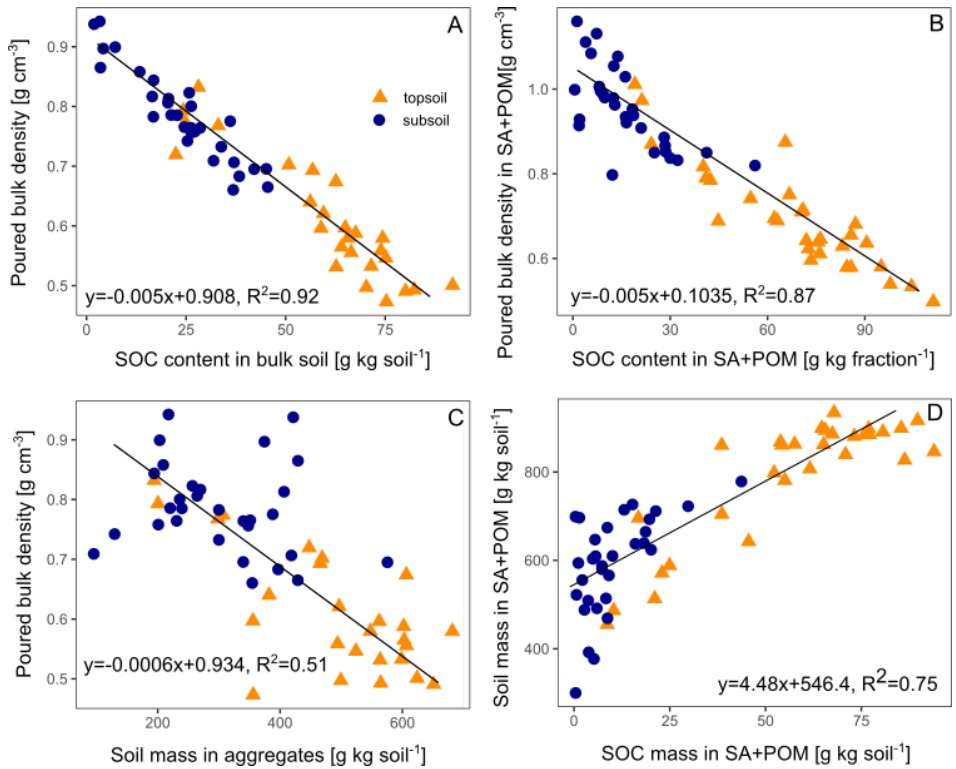

Figure 6: Poured bulk density as a function of soil organic carbon (SOC) content in A) the bulk soil and B) the coarse (>63 µm) soil fraction (sand and stable aggregates=SA and particulate organic matter=POM); C) poured bulk density as a function of soil mass in aggregates and D) soil mass in the coarse soil fraction as a function of SOC mass in the coarse soil fraction with regression models fitted to all observations.