# Peer review of "Strong warming of a subarctic Andosol depleted soil carbon and aggregation under forest and grassland cover"

_SOIL, 2019_

## Short Comment (SC1) · 25 Aug 2019

Dear Christopher, Páll and Bjarni

It was a pleasure to me to find a manuscript which deals with a very similar fraction-ation approach and which presents an Icelandic study. Hence, it is very similar to my research which I did in Iceland some years ago. My memories from the southern part of Iceland came back... Therefore, I read the manuscript of your study and would like to give you my comments.

1) It is ingenious to use a natural heat source and the resulting warming gradient to

study ecosystem changes. This is likely possible on a volcanic island like Iceland. But on volcanic islands, soil properties differ from other soil types of non-volcanic regions in the boreal ecosystem. How much are your results applicable to the rest of the boreal ecosystem? What could be the limitations?

2) You mentioned that the worst scenario in the IPCC report predicts a temperature increase of 11 °C for the region North of 60°N. Your temperature sequence ends at +17.5°C. Can you link your results to the different IPCC scenarios with regards to the maximum predicted temperature increases in the IPCC report? Can you make any assumption, how the SOC change would look like at +11°C (IPCC report) based on your temperature sequence (..., 5.8°C, 17.5°C). By the way, does the referenced temperature increase (+11°C) corresponds to the air temperature? In this case, how do you link the increase of air temperature to the increase of soil temperature? Does soil temperature increase in the same way and with the same slope gradient in the future and therefore does your soil warming gradient corresponds to the assumed soil temperature increase in the future for boreal ecosystems? Please clarify this in the introduction part.

3) Based on Comment 2, could you make any statements about the impact of tem-perature increase on SOC regarding the different IPCC scenarios (e.g. SOC change according to the smallest temperature increase).

4) In your study the temperature increased within 10 years. The modelled increase of air temperature will however change within several decades. What would you think, is the different time scale irrelevant concerning the change of SOC and the soil processes which control the SOC?

5) Is the vegetation (grassland or forest ecosystem) also changing during the warming within these 10 years? If there is also a change in the vegetation composition or the supply of OM to the soil phases, might these changes also be responsible for the changes in your SOC results? In this case, the increase of the temperature is not the

only independent parameter that changes.

6) Discussing the change of soil structure and SOC content (within SOC fraction), you might also need to have a look at the soil mineralogy. Did you analyse the volcanic clay minerals? Volcanic clay minerals and the abundance or change of metal-humus complexes, allophane content, ferrihydrite content can also explain the changes within more resistant SOC fractions.

7) What do you suggest to use as a further analysis technique to characterize the stable SOC fraction (rSOC) or the <63 microns fraction? I ask you, because I used the same fractionation technique and later, however, read that the wet oxidation step is questioned. Spontaneously, I would measure the SOC and the volcanic clay minerals in the <63 microns fraction to get an idea about the characteristics between the SOC and the soil mineralogy. Do you have any further ideas which approaches can be used when the SOC of volcanic soils is fractioned and characterized?

Statistics 8) How many replicates do you have per category (e.g. Fig. 4)? Is it n=5? Can you mention the number of samples per category and the number of samples within the two ecosystem datasets in section 2? It might be also useful to mention it in the capture of Table 1.

9) Mention also p-values in the text when giving the regression values (e.g. page 6 line 31, Figure 6). I remember the reviews of my manuscript. . .

10) In Figure 4 and Figure 5, what does the intervals along the regression lines indicate? Is it the 95% range of the regression value or the 95% range of the modelled value?

11) Changes of ecosystem processes are not always linear. Figure 4 and Figure 5 show distributions of values which could be modelled more accurate with a non-linear function. For example, the patterns of the contents of bulk SOC, SA, POM or rSOC show different slopes along the temperature gradient and asymptotic properties. Did

you tested other types of functions to explain the patterns of SOC changes? There might be also a non-linear correlation in Figure 6 D.

12) In the case of a non-linear pattern, is it useful to show only one absolute change value (g C kg-1 fraction $°C-1$)?

13) In the case of a non-linear pattern, when do you expect the highest rates of change? Is this in the beginning of the warming or at the end? What does this mean for the change of the boreal ecosystem and at what time do you expect the highest changes (in the next 10-20 years or in 80-100 years)?

14) I offer you to read the reviews of my manuscript. The study deals also about SOC fractions in volcanic soils in Iceland and some comments might be also useful for the revision of your manuscript. https://www.soil-journal.net/5/223/2019/soil-5-223-2019-discussion.html

15) In this journal, square brackets are not used to note any units (e.g. [$°C$]). It uses parenthesis ($°C$).

16) I guess that Figure 5 shows the scatterplots for the SOC contents in the subsoil (20-30 cm). Please change the title in the Y-axis.

Basel, 25th August 2019 Matthias Hunziker

---

## Referee Comment (RC1) · Anonymous Referee #1 · 1 Oct 2019

The comment was uploaded in the form of a supplement:
https://www.soil-discuss.net/soil-2019-41/soil-2019-41-RC1-supplement.pdf
* * *

---

## Referee Comment (RC2) · Anonymous Referee #2 · 14 Oct 2019

General comments:

The manuscript by Poeplau et al., investigates the effects of long-term soil warming on SOC fractions. Their carbon content as well as their relative distribution in response to warming are being discussed for two different soil depths and ecosystems. As I myself am working on warming effects on soil microbial communities in connection to biogeochemical cycles, I was very pleased to read about warming effects on the abiotic components of soil. Especially long-term in-situ warming experiments are rare and extremely valuable to study the mechanisms and concepts behind various warming effects. I completely agree with the authors that strong systematic gradients in SOC

content (in the same soil) can provide an important framework to improve our understanding of SOC dynamics. I believe that this study adds some very valuable aspects the research field of soil warming. This paper and some there presented ideas could also provide the basis for hypotheses that could be targeted in other future mechanistic studies. The manuscript is well written and structured. Most parts of it are easy to follow, however I hereby want to suggest some minor and detailed revisions in order to improve the manuscript in terms of its readability and understandability.

Personal comments:

Out of personal interest, I would like to ask if you have a suggestion to why both ecosystem types approach to more or less the same SOC content in response to long-term warming (Fig. 4 and 5)? The levelling off of (absolute) SOC losses from different soil types (with different native C contents) to a sort of threshold level is very interesting to me.

I also like the idea of decreasing pore space with warming (due to aggregation loss). I myself often observed a decrease in microbial biomass with warming, which I now start considering to link to the suggested decrease in pore space and microhabitats. Therefore, one can see the need of better exploring structural soil changes upon warming in order to better interpret biotic responses.

Out of curiosity I also want to ask if you observed any changes in the vegetation cover after several years of soil warming? And if yes, if there is data on plant communities available or a paper covering that aspect? As far as I know, changes in vegetation with warming are quite often observed and I wonder about associated impacts on SOC contents and fraction distribution. E.g. if observed effects like the loss of aggregation could be associated with a change in the vegetation structure?

Out of personal interest, is the light ultrasonic treatment more effective and more representative for natural conditions than the slacking treatment?

Specific comments:

Short comment to the title: I am not sure if you can really call it a "deteriorated" soil structure. I would rather use the term "changed" or "affected" because it does not imply a judgement. Also. The authors sometimes refer to boreal systems throughout their manuscript. However, I much better like the term subarctic in the context of the study site. Throughout the manuscript, I found an inconsistency in terms of SOC terminology as it is stated on page four and also within the use of associated units. In order to improve the readability and the understandability of the text, I would also welcome to be more specific and explicit when writing about SOC changes e.g. SOC contents, concentrations, fractions, mass et cetera. For detailed comments about alternative phrasing suggestions see below.

Page 1:

Line 14: Five different SOC fractions were isolated and their re-distribution as well as the amount of stable aggregates was assessed to link SOC to soil structure changes.

Line 16: Soil warming had depleted SOC concentrations in forest bulk soil by...

Line 24: ...indicating an indirect protective effect of SOC on aggregates...

Line 25: Topsoil changes in total SOC content and fraction distribution...

Line 27: ...in the response of subsoil SOC content and fraction distribution... The authors write in the abstract that no ecosystem effect was observed. However, this was confusing to me, as in my understanding Tab. 3 shows significant effects.

Line 32: Could you please specify if the stated temperature increase refers to air or soil temperature?

Line 3: I believe that the statement about permafrost soils is out of place here as no permafrost is occurring at the investigated site.

Line 32: It was a bit hard to understand to how many samples you refer in your manuscript. Could you please state the explicit number of samples taken? Also, are all mentioned five transects situated in the investigated forest? Please also indicate the number of samples in your graphs.

Line 6: Distinct responses to warming were thus expected. Could you make an explicit statement what you were expecting?

Line 15: Unit is missing for SPT (1.8 g cm3 -1)

Line 33: Out of personal interest - what is the variability to the average mass recovery?

Line 11: Would you assume a positive or negative correlation between the poured bulk density and SOC in the SA fraction?

Line 31: Do you think that absolute SOC losses are higher in topsoil because of a higher "native" C concentration?

Line 1: Please state here that you talk about SOC contents in bulk soil.

Line 11: The depletion of SOC content lead to a changed relative distribution ...

Line 12: The ANOSIM revealed... Please state here that you talk about topsoil findings.

Line 13: ...fraction distribution was only significant from the unwarmed reference at a warming intensity of 5.8°C. I was wondering about the + 2.7 °C treatment (see Tab. 2)?

Line 15: ...SOC in the POM and SA fractions, which were strongly depleted with warming (Fig 1). Please mention here, that this was the case in both depth increments.

Line 28: . . .the relative mass proportion of rSOC was expected to increase. . .

Line 30: Could you please give the p-value for the mentioned regression between rSOC and total SOC in the SC fraction?

Line 36: Could you please state the p-value for the significant negative relationship between the proportion of SOC in SA and the proportion of SOC in DOC

Line 5: You mention similar SOC contents for subsoil forest and grassland soils. Is there more information about that e.g. an ANOVA?

Line 14: You mention that POM in forest soils responded more negatively to warming than POM in grassland soils. Was this normalized to their respective C contents?

Line 17: I would have appreciated it, if you mentioned earlier on that the warming in the grassland soils was only 6 years compared to the 10 years of warming in the forest.

Line 20: . . .we found a strong negative correlation of bulk soil SOC content and poured BD. Please also give the R2 here.

Line 34: According to Tab. 1 the relative change in topsoil SOC content is -3.6% not -2.7% as stated.

Line 36: Do you think that the + 5.8°C is a realistic warming intensity for soil or air temperatures?

Line 17: You write, that the present study did not reveal tipping points. However, if I look at e.g. Fig 1. it seems to me that +5.8°C causes some abrupt changes in SOC contents and SOC proportions in fractions? Please also specify "tipping points for SOC" here (e.g. SOC contents).

Line 22: You write that climate change is likely to strongly affect SOC stocks of boreal

forests. Generally, I strongly agree to that statement. However, would question the comparability of the relatively young investigated forest on volcanic bedrock material to the biome of naturally old grown boreal forests. Upscaling to the regional or global context might be a slight over interpretation.

Line 17: In the unwarmed reference soil, it accounted for the highest proportion of soil mass and SOC content.

Line 29: . . .we found a very strong positive correlation of SOC mass and . . .

Line 2: I did not understand the context of the sentence about carbon desorption from the mineral phase. It seemed a bit out of place to me.

Line 8: According to your definition on page 4 the unit of SOC content should be (g C kg -1).

Line 7: You might rephrase the sentence to: Changes in SOC concentrations and the relative distribution of fraction masses in the grassland soils have been previously investigated.

Line 8 to Line 11: The fact that there is no difference in subsoil SOC dynamics . . . might indicate that the same mechanisms of SOC depletion were involved in both ecosystems.

Line 23: The sentence is very long, maybe you could split it apart.

Line 37: You might rephrase the sentence to: Differences in the relative distribution of SOC fractions and their respective SOC concentration in response to warming have only been found in the topsoils of both examined ecosystems.

Specific comments about graphs and tables:

1) Tab. 2: In the table description you write about testing "differences in SOC fraction distribution". Do you mean the relative mass distribution of SOC fractions?

2) Fig.1: is missing the a)b)c)d) notation in the individual graphs. Also the unit of the x-axis of a) and b) should be changed to SOC content (g C kg-1).

3) Fig.3: For me, the graph would be easier to understand if the title of the x and y axis would be changed to "percentage of total SOC in SA" and "percentage of total SOC in DOC". Also a p-value is missing in the graph as what is the depicted error range (95% confidence interval?).

4) Fig 4: I am a bit confused if the scatter plots show SOC masses or SOC contents of the fractions in response to warming. What is the depicted error range? Moreover, some relationships seem rather curvilinear to me than linear.

5) Fig 5: Please change the title of the the y axis to subsoil instead of topsoil. I moreover have the same small issues with the graphs as mentioned for Fig. 4.

6) Fig. 6 shows regression models. Please indicate the p-values here. 6a) The unit of SOC content should be (g C kg soil -1). 6b) The unit of SOC concentration should be (g C kg fraction -1). 6c) Please change the title of the x axis to " soil mass in stable aggregates". 6d) The shown relationship looks more curvilinear than linear to me.

Questions provided by SOIL:

1) Does the paper address relevant scientific questions within the scope of SOIL? Yes, I think so.

2) Does the paper present novel concepts, ideas, tools, or data? The paper shows new interesting data on physical soil structure changes in responses to warming. However, to my state of knowledge, no new tools were involved. The manuscript provides some new concepts and ideas e.g. the proposed mechanism of warming leading to SOC loss

(via enhanced microbial activity) which then results in the loss of stable aggregates. I also very much appreciate the proposed idea that the slope of the regression line between SOC and bulk density might be a useful indicator for aggregation affinity in unmanaged soils.

3) Does the paper address soils within a multidisciplinary context? n.a.

4) Is the paper of broad international interest? The scope of physical soil fractions and their response to warming seems of broad interest. It represents a framework of many biological responses to higher temperatures.

5) Are clear objectives and/or hypotheses put forward? Three objectives are stated clearly on page three and then also addressed in the results and discussion of the paper.

6) Are the scientific methods valid and clear outlined to be reproduced? Yes. Especially the SOC fractionation protocol is described in great detail and could be repeated in our lab too.

7) Is the soil type/classification adequately described? Yes, most of the general information on soil types is given in the text.

8) Are analyses and assumptions valid? Yes.

9) Are the presented results sufficient to support the interpretations and associated discussion? Yes.

10) Is the discussion relevant and backed up? In general yes. For detailed comments see above.

11) Are accurate conclusions reached based on the presented results and discussion? Yes.

12) Do the authors give proper credit to related and relevant work and clearly indicate their own original contribution? Yes.

13) Does the title clearly reflect the contents of the paper and is it informative? Yes. The title clearly reflects the later proposed mechanism of aggregate break-down which follows SOC loss that was caused by warming. I especially like that the title includes the term "subarctic" and not "boreal".

14) Does the abstract provide a concise and complete summary, including quantitative results? Yes.

15) Is the overall presentation well structured? Yes, I like the tripartite structure of the paper (1-warming effects on forest SOC and its fractions, 2-forest vs. soil SOC in response to warming, 3- soil structural changes). The focus on those three topics can be found in the introduction, results and discussion part.

16) Is the paper written concisely and to the point? To my understanding the manuscript is mostly concise. However, sometimes the sentences were hard to follow (too long) and not precise enough to understand what the authors meant. This holds especially for "SOC-terminology and SOC units" – see detailed comments above).

17) Is the language fluent, precise, and grammatically correct? Mostly yes. However, some sentences are relatively long and thus hard to follow. This is especially the case in the discussion part.

18) Are the figures and tables useful and all necessary? The figures are nice and useful. For detailed suggestions see comments above.

19) Are mathematical formulae, symbols, abbreviations, and units correctly defined and used according to the author guidelines? Yes. In the context of units, please see detailed comments above.

20) Should any parts of the paper (text, formulae, figures, tables) be clarified, reduced, combined, or eliminated? I think that the sampling procedure (amount of taken samples and analyzed) could be described in more detail.

21) Are the number and quality of references appropriate? Yes.

22) Is the amount and quality of supplementary material appropriate and of added value? Yes.

---

## Author Comment (AC1) · 13 Nov 2019

Matthias Hunziker matthew_hunziker@gmx.ch

Dear Christopher, Páll and Bjarni It was a pleasure to me to find a manuscript which deals with a very similar fraction- ation approach and which presents an Icelandic study. Hence, it is very similar to my research which I did in Iceland some years ago. My memories from the southern part of Iceland came back... Therefore, I read the manuscript of your study and would like to give you my comments.

1) It is ingenious to use a natural heat source and the resulting warming gradient to study ecosystem changes. This is likely possible on a volcanic island like Iceland. But on volcanic islands, soil properties differ from other soil types of non-volcanic regions in the boreal ecosystem. How much are your results applicable to the rest of the boreal ecosystem? What could be the limitations?

Answer: Each case study has its limitations and should not be extrapolated to whole biomes. This is true when talking about absolute rates of change, while the direction of change or certain response mechanisms might well be more generic. The soil of this study is a quite specific one, and we agree that we have not accounted for this enough in the previous version. We now mention the word Andosol in the title to clarify the focus.

2) You mentioned that the worst scenario in the IPCC report predicts a temperature increase of 11°C for the region North of 60°N. Your temperature sequence ends at +17.5°C. Can you link your results to the different IPCC scenarios with regards to the maximum predicted temperature increases in the IPCC report? Can you make any assumption, how the SOC change would look like at +11°C (IPCC report) based on your temperature sequence (5.8°C, 17.5°C). By the way, does the referenced temperature increase (+11°C) corresponds to the air temperature? In this case, how do you link the increase of air temperature to the increase of soil temperature? Does soil temperature increase in the same way and with the same slope gradient in the future and therefore does your soil warming gradient corresponds to the assumed soil temperature increase in the future for boreal ecosystems? Please clarify this in the introduction part.

Answer: This comment includes a few important remarks at the same time. Of course, soil warming is not equal to air warming. The difference between soil temperature and air temperature is site specific (surface energy balance) and therefore it is hard to predict how much air temperature increase is necessary to increase the average soil temperature e.g. by 5.8°C at a specific site. Also it is clear that the source of warming may have a strong effect on the overall ecosystem response. However, based on the

most pessimistic IPCC extreme scenarios, we think that at least a soil warming of 5.8°C is within this (maximum) realistic range. We have now been clearer in the discussion when we talk about warming if we refer to air temperature or soil temperature (also because one reviewer requested that). However, the introduction is not the place to discuss shortcomings of geothermal warming and also not how we can link our results with any IPCC scenario or how soil temperature will increase in the future. There is a lot of uncertainty involved, especially regarding the long-term effects of soil warming. And this is the reason why we mention (in the introduction) that long-term warming experiments are needed. Also, the warming response of bulk SOC was linear, so it is easy to derive a number for any of the suggested warming scenarios, if you wish so - with high uncertainties: only one soil, soil warming not air warming, warming from below, very abrupt warming vs. long term gradual warming... Thus, we do not want to speculate too much and go too far into this direction and see the strength of this experiment more in the fact that we have strong and long soil warming to study mechanisms.

3) Based on Comment 2, could you make any statements about the impact of temperature increase on SOC regarding the different IPCC scenarios (e.g. SOC change according to the smallest temperature increase).

Answer: For mentioned reasons (in answers 1 and 2): No. Recent papers (e.g. Crowther et al.) tried to predict changes in SOC until the end of the century or until 2050 based on short-term warming experiments. We think that this is problematic for several reasons and do not dare to directly conclude from our still relatively short-term warming experiments what will happen under global change in the long run.

4) In your study the temperature increased within 10 years. The modelled increase of air temperature will however change within several decades. What would you think, is the different time scale irrelevant concerning the change of SOC and the soil processes which control the SOC?

Answer: This is a good comment and it concerns basically all climate change manipulation experiments (temperature, CO2 conc., drought, ozone, UV, etc.) that have been conducted globally. No, it is surely not irrelevant. We might see an overshoot-reaction here, or things might level off soon. We have indications from different ForHot papers that such responses can occur for when the 10 year warming is compared to >50 year warming in the same grassland type. This is actually the focus of a Nature Ecology and Evolution paper on that exact aspect soon where a large range of ecosystem parameters are compared (Walker et al., in press). The main message is: Be careful with linear extrapolations of short-term warming experiments. The doubts that you mention in your last 4 comments are all correct, but I think at the same time it is pretty clear that our experiment cannot solve the equation for all the boreal zone, all IPCC scenarios and until the end of the century. To clarify this, we added the following sentence to the discussion (p8, l.37ff: "However, the transferability of the results in this study to the SOC response to global warming is still rather limited and can only slightly reduce given uncertainties: i) we studied soil temperature, not air temperature increase, ii) the warming occurred abruptly and not gradually, iii) we studied an Andosol. Extrapolations to larger areas or longer time periods should thus be done carefully and were not intended with this study."

5) Is the vegetation (grassland or forest ecosystem) also changing during the warming within these 10 years? If there is also a change in the vegetation composition or the supply of OM to the soil phases, might these changes also be responsible for the changes in your SOC results? In this case, the increase of the temperature is not the only independent parameter that changes.

Answer: Yes, there were some changes in vegetation, while those were most pronounced in the most extreme warming intensities >15°C: For example, in the forest many trees died after this extreme and abrupt soil warming event. However, the changes in SOC we see are very gradual (along the temperature rise, so this is not reflected in the SOC data). Also, there is more understorey herbaceous vegetation

now in the warmest treatment due to the canopy reduction, so the input of C might not even be smaller. However, for both ecosystems we actually detected a decrease in root biomass and (although less pronounced) also in litter biomass. There are ongoing studies on if the turnover rates of those roots also change, but those are still unpublished. It is thus quite realistic, that changes in SOC are also partly driven by decreased C input, although biomass turnover is not comprehensively measured yet. So yes, temperature increase is the independent parameter that changed in the very beginning, but then there is a whole chain of mechanisms that could lead to changes in SOC and SOC fraction distribution. We don't go into too much detail here, because that was not the focus of the study and data availability is partly scattered yet. However, we added the following sentences (in section 4.1): "In fact, root biomass in 0-10 cm decreased in both ecosystems (data not shown), leading to weak positive correlations ($R^2$=0.37 for forest and $R^2$=0.29 for grasslands) of SOC and root biomass. Also aboveground plant litter tended to decline in both ecosystems. This suggests that SOC losses were partly driven by decreasing C input with warming and not by increased microbial activity alone. However, a clear picture on absolute C inputs in the experimental plots is not available yet, since it needs to consider NPP and biomass turnover at the same time."

6) Discussing the change of soil structure and SOC content (within SOC fraction), you might also need to have a look at the soil mineralogy. Did you analyse the volcanic clay minerals? Volcanic clay minerals and the abundance or change of metal-humus complexes, allophane content, ferrihydrite content can also explain the changes within more resistant SOC fractions.

Answer: This would surely be an interesting thing to do, especially for a better characterization of the soil, but was not done in this study. We acknowledge that those Andosol-specific characteristics are important, but we think that i) temperature changes were not high enough to change soil mineralogy substantially and ii) mineralogy is more related to SOC stabilization mechanisms in the finest fractions (including microaggregates), while here we observe the strongest changes in the larger fractions. We believe that the response that we observe (i.e. a break-down of large micro- and macroaggregates) is more controlled by biotic than by abiotic drivers. Organo-mineral interactions and aggregates in Andosols are known to be particularly strong and almost impossible to disintegrate, but what we see here is a break-down of aggregates due to warming. It seems unrealistic, that this related to the specific mineralogical features of an Andosol. But we agree that further studies should include analyses that can be used to quantify metal-humus complexes, allophane content, ferrihydrite. The plan is to get your friend prof. Ólafur Arnalds involved in the near future to add those aspects to a further study.

7) What do you suggest to use as a further analysis technique to characterize the stable SOC fraction (rSOC) or the <63 microns fraction? I ask you, because I used the same fractionation technique and later, however, read that the wet oxidation step is questioned. Spontaneously, I would measure the SOC and the volcanic clay minerals in the <63 microns fraction to get an idea about the characteristics between the SOC and the soil mineralogy. Do you have any further ideas which approaches can be used when the SOC of volcanic soils is fractioned and characterized?

Answer: There are different ways of wet oxidation and the one used in this study (with NaOCl) was found to be related (in previous studies) to the amount of extractable Fe and Al in the soil, so directly to soil mineralogy, which makes sense. It seems to be those organo-metal complexes that resist that oxidation step and that form the most stable SOC. The derived fraction rSOC is mostly found to be the one that shows the least response to any change or novelty, therefore we think that wet oxidation (especially with NaOCl) gives meaningful results to some extent. But, indeed it remains a mystery why this is not the case for warming (or at least in this experiment): rSOC shows pretty much the same response as the silt and clay fraction as a whole. If this is related to the type of soil or the type of treatment (warming) remains open. Adding mineralogy data to fraction data could always be helpful, but not sure if it will help to solve that specific question. Thermo-stability, as assessed using techniques like Rock-Eval

can also be promising to determine the amount of persistent SOC- with its own methodological difficulties. We have slightly changed the section on this fraction in section 4.1: "This has been observed before and questions the notion that this oxidation-resistant pool can be linked to a centennially persistent or even inert SOC pool (Lutfalla et al., 2014;Poeplau et al., 2019;Poeplau et al., 2017;Zimmermann et al., 2007). At the same time, NaOCl-resistant SOC has often been described as substantially older and thus slower cycling as bulk SOC (Helfrich et al., 2007) and was also found to correlate to the abundance of Al and Fe-oxides in the soil (Mikutta et al., 2005). Thus, the strong warming response of this fraction is somewhat in contrast to the slow responses observed to other treatments, such as C3-C4 vegetation changes (Poeplau et al. 2018)." Additional comment from Bjarni: We recently got some 14C analyses from the topsoil and subsoil and also from the <63 microns fraction in the grassland sites. It unexpectedly showed only minor age differences... For me this seems to indicate that in our soil/ecosystem we are having a very high turnover of SOC, which is also in line with what we are finding... Again, this will be addressed in more detail in future paper, where we will look more closely into those dynamics and also involving the mineralogy.

Statistics 8) How many replicates do you have per category (e.g. Fig. 4)? Is it n=5? Can you mention the number of samples per category and the number of samples within the two ecosystem datasets in section 2? It might be also useful to mention it in the capture of Table 1.

Answer: It is 5 and we have now entered another sentence on that in the soil sampling section (although it was already mentioned before in the experimental design section). Also in the table capture of Tab.1 we included it now, thank you.

9) Mention also p-values in the text when giving the regression values (e.g. page 6 line 31, Figure 6). I remember the reviews of my manuscript

Answer: Done.

10) In Figure 4 and Figure 5, what does the intervals along the regression lines indicate? Is it the 95% range of the regression value or the 95% range of the modelled value?

Answer: It is the 95% confidence interval of the regression model- this information was added to all Figure captions.

11) Changes of ecosystem processes are not always linear. Figure 4 and Figure 5 show distributions of values which could be modelled more accurate with a non-linear function. For example, the patterns of the contents of bulk SOC, SA, POM or rSOC show different slopes along the temperature gradient and asymptotic properties. Did you tested other types of functions to explain the patterns of SOC changes? There might be also a non-linear correlation in Figure 6 D.

Answer: We admit that it was a bit oversimplified to assume linearity in all cases. We now tried logarithmic fits as a second option and indeed, it did fit better in several cases. Figure 4 and 5 were adjusted and so was the Statistics section. However, in Figure 6, a linear fit was much better – the optical non-linearity might originate from the three points at the lower left corner, while there is a wealth of observations much higher than those. We therefore did not adjust Figure 6.

12) In the case of a non-linear pattern, is it useful to show only one absolute change value (g C kg-1 fraction°C-1)?

Answer: We agree that using this one value is again oversimplified in case of a clear non-linear response. However, i) the values used in the abstract (and directly in the text) refer to linear responses of topsoil and subsoil bulk SOC and ii) the values given in table 1 (for the fractions) are used as measures to directly compare the response of different fractions. In this sense, we think it is useful to give also these values. The only other chance we see is to use the difference between the unwarmed reference and the highest warming, but we think that a linear fit is a more fair comparison, since it does not overemphasize those two values. We have now added the following sentence to the table heading of table 1: "Although this was not the best model in all cases, we

used this value as a proxy to compare the warming response among fractions."

13) In the case of a non-linear pattern, when do you expect the highest rates of change? Is this in the beginning of the warming or at the end? What does this mean for the change of the boreal ecosystem and at what time do you expect the highest changes (in the next 10-20 years or in 80-100 years)?

Answer: This is a different question: We looked at the response to a temperature gradient; you are now referring to the temporal pattern. This is a different story and very uncertain (as mentioned earlier). The dataset investigated here cannot be used to answer this important question, but we might have indications that the loss actually levels off after the first years (mentioned in the discussion) – and the paper on this issue involving many different ecosystem processes and characteristics is in press in Nature Ecology and Evolution (Walker et al.)

14) I offer you to read the reviews of my manuscript. The study deals also about SOC fractions in volcanic soils in Iceland and some comments might be also useful for the revision of your manuscript. https://www.soil-journal.net/5/223/2019/soil-5-223-2019-discussion.html

Answer: Thank you.

15) In this journal, square brackets are not used to note any units (e.g. [°C]). It uses parenthesis (°C).

Answer: Thank you, we have changed it everywhere.

16) I guess that Figure 5 shows the scatterplots for the SOC contents in the subsoil (20-30 cm). Please change the title in the Y-axis.

Answer: This is correct and was changed accordingly.

---

## Author Comment (AC2) · 13 Nov 2019

In large this is a good and interesting paper. However, there are some shortcomings with regard to these types of studies; The authors themselves have discussed problems connected to interpretation of results from experiments using geothermically warmed soils in a global warming / climate change context, concerns that I also share. Though I am not convinced that the study is bringing us much forward in questions regarding the fate of carbon in subarctic forest soils in future warmer climates, I still see the relevance with regard to the fate of soil carbon in geothermically warmed Andesitic soil. I believe the most important results in this study are the relative changes in proportion of C in

the different fraction with increasing temperature and this should be more in focus than the loss of C and deterioration of soil structure. I also think that the differences between the two ecosystems (grassland and forest) should be better communicated in the tittle.

Answer: We thank the reviewer for the valuable comments and suggestions. The title has been adjusted to i) narrow the focus to Andosols and ii) include grasslands. Regarding the focus, we see that loss of C, fraction distribution and soil structure are closely coupled in this case and we tried to make exactly this point in the study.

More specific comments

1. Does the paper address relevant scientific questions within the scope of SOIL? yes

2.Does the paper present novel concepts, ideas, tools, or data? the paper presents new date, but is not particularly novel in concept, ideas and tools.

3. Does the paper address soils within a multidisciplinary context? Yes, in a using geothermic warming of soils as a proxy for warming of soils in climate change scenario. But does not address the ecosystem changes/ vegetation as much it ought

4. Is the paper of broad international interest? Yes, but not as broad as the title suggests, these are Andesitic soil and thermal warming of soils do have some limitations with regard to interpretation in a global change context.

5. Are clear objectives and/or hypotheses put forward? Yes, three objectives are stated. 1.advance our understanding of the temperature response of different SOC fractions representing kinetic pools 2.assess the role of the ecosystem type in the temperature response of SOC 3.investigate potential links between SOC loss and so il structure changes.

6.Are the scientific methods valid and clear outlined to be reproduced? I have some questions with regard to sampling and interpretation of the term soil structure, see below.

SOILD
7.Is the soil type/classification adequately described? fairly, general information on soil type/classification at the experimental site is given. but I cannot see that the information that the soil type/classification provides is actually used in the interpretation of the results. Though only the upper 30 cm is used in this study it would have given valuable information if this was related to soil horizons.

Answer: The whole study design was not related to genetic horizons, but to fixed depth increments, which is a standard e.g. when different ecosystems are compared.

8.Are analyses and assumptions valid? see comments below

9.Are the presented results sufficient to support the interpretations and associated discussion? see comments below

10.Is the discussion relevant and backed up? see comments below

11.Are accurate conclusions reached based on the presented results and discussion? Yes

12.Do the authors give proper credit to related and relevant work and clearly indicate their own original contribution? Yes

13.Does the title clearly reflect the contents of the paper and is it informative? See comments below

14.Does the abstract provide a concise and complete summary, including quantitative results? Yes

15. Is the overall presentation well structured? yes

16.Is the paper written concisely and to the point? OK

17.Is the language fluent, precise, and grammatically correct? long cumbersome sentences make particularly the discussion, but also elsewhere. You are really not making it easy for the reader when you write sentences like this e.g.: Page 9 lines 30 to 33 SOILD
" Thereby, topsoil and subsoil samples scattered approximately around the same regression line, indicating that especially the amount of young SOC, may it be driven by warming intensity or the location within the soil profile, was a very good predictor for the amount of stable aggregates in the soil." Or page 11 line 24-26: "Together with the fact that also more labile SOC was present in the forest topsoil, which responded more sensitive to warming than the soil, it seems likely that amount and fraction distribution of SOC drove the ecosystem specific warming response in the topsoil."

Answer: We agree that some sentences, especially in the discussion, were too long and reworked those. The mentioned sentences now read as follows: "Topsoil and subsoil samples scattered approximately around the same regression line. This indicates that the abundance of young and coarse SOC per se, rather than the degree of soil warming, is driving the amount of stable aggregates in the soil." And "Therefore it seems likely that amount and fraction distribution of SOC drove the ecosystem specific warming response in the topsoil." We further changed the sentence "In consequence of SOC loss, total pore space decreased strongly as indicated by poured bulk density used as a proxy for bulk density in undisturbed samples. The latter was unfortunately not determined in the present study." Into "In consequence of SOC loss, total pore space decreased strongly as indicated by poured bulk density. Poured bulk density was used as a proxy for in situ bulk density in the undisturbed soil, which was unfortunately not determined in the present study." And "While the authors did not explicitly link that to changes in soil structure, an increase in bulk density with associated decrease in pore space might have fostered that change." Into "An increase in bulk density with associated decrease in pore space might have fostered this physiological response, although this was not explicitly mentioned by the authors."

18. Are the figures and tables useful and all necessary? Yes (see point 20)

19. Are mathematical formulae, symbols, abbreviations, and units correctly defined and used according to the author guidelines? I believe so
20.Should any parts of the paper (text, formulae, figures, tables) be clarified, reduced, combined, or eliminated? 1.most of the text is relevant and to the point, but I am unsure if the "poured bulk density - soil structure" part contributes in a meaningful way.D eleting it and giving more focus to the warming effects on fractionation -ecosystem comparisons stable soil C would benefit the paper.

Answer: We do not necessarily agree on that point, and for example reviewer 2 was especially in favour of that part. We therefore decided to keep the focus as it is, although we clearly acknowledge that the proxies used are not enough to fully understand the SOC effect on soil structure in the studied soil.

2.Most of the figures are nice and useful. Figure 5 must have the wrong Y-axis tittle – subsoil not topsoil SOC. However, I wonder if it would be possible to add the warming (some sort of colour code) to Figure6. It would be interesting to see if there were any systematic also with regard to temperature.

Answer: Figure 5 has been changed. In Figure 6, we think that giving the different warming intensities would over-complicate the figure. It already consists of 4 panels and thus a high information density. In the plots before, we show how warming and SOC are correlated, so this would be redundant information to some degree.

3.1 miss a table showing general soil properties such as pH, oxide extraction of some sort (Fe, Al, Si), sand silt clay%. I am not too keen on table 2 and 3 which only show summary of statistics, they are much more valuable when they are connected to measured properties, merge or delete?

Answer: Soil pH, texture and initial SOC contents are given in the text describing the experiment. Fe, AI and Si contents were not measured. Also, some more information, e.g. on soil temperature profiles, are given in the mentioned publication which is a pure site description publication. Therefore, we think that an additional table is not necessary. We also think that both, table 2 and 3 are necessary and would like to keep them.

SOILD
21. Are the number and quality of references appropriate? Yes

22. Is the amount and quality of supplementary material appropriate and of added value? as far as I can see yes

I have a couple of more specific comments that I think the authors need to address to improve the paper

1)I think the use of the term "soil structure" is used wrong. Soil structure has not been investigated in this study. What has been studied is stable aggregate (SA) (63-2000 $\mu$ m) from the fractionation procedure and the carbon (C) connected to this fraction. The natural soil bulk density (BD) was not measured – which could have given some indication of soil structure and the "poured bulk density" does not replace the measurement of the natural BD, though it does give a relative difference between dried soil material in the fraction less than 2 mm. I therefore suggest a change in tittle. e.g.: Âń Strong warming of subarctic forest soil reduces stabilisation of carbon in soil aggregates – Indications from organic matter fractionation". I would also suggest including" subarctic andesitic forest soil" in the tittle as these soils normally show both chemical and physical properties that are markedly different from "subarctic forest soils" formed on other parent materials.

Answer: We agree that using the word soil structure in the title is misleading, or leads to different expectations. We also agree that the title was too brought, since we investigated a very specific soil type. Therefore we changed the title to: 'Strong warming of a subarctic Andosol depleted soil carbon and aggregation under forest and grassland cover'.

2) Sampling method and comparisons, sampling should be better explained, and it would also be nice to know how many samples (N=) were behind each temperature/location? In studies like this sampling method and a good description of these are crucial. Many sophisticated analyses in the laboratory will never compensate for errors, flaws and inaccuracy in sampling- or description of sampling. I am afraid that

SOILD
the lack attention to the sampling procedure maymake the results of this study none reproducible. Personal experience with similar sampling schemes suggests that a soil of this nature (Silandic Andosol with a silt loam texture, 60 % silt) is easily compressed during sampling. How were the depth intervals determined? If sampling was done as I anticipate by extracting a 30 cm cylindrical core and then splitting it into 0-10, 10-20 and 20-30 samplesthis could be a real challenge. The comparisons between the different layers could be based on pure artefacts - please convince me of the opposite

Answer: The amount of samples has been introduced in the section 2.1, which was renamed from study site to study site and experimental design. However, we also added the number of warming intensities and replicates to the section 2.2 and the sentence now reads as follows:'In late April 2018, i.e. almost exactly 10 years after the warming was initiated, mineral soils of all permanent forest plots (six warming intensities, five replicates each) were sampled.' Regarding the sampling and potential artefacts: We used a thin auger that is not necessarily suited for volumetric sampling, but has the advantage that soil compaction is relatively low. However, as the reviewer mentioned correctly, this kind of soil is easily compressed. This is especially true for the warmer treatments that are less stabile due to less SOC and less aggregates. Changes in soil structure and structural stability were thus already noticed during sampling. The procedure was the following: The auger was always driven into the soil to a depth of 30cm. When extracting the auger with the respective soil core, the soil core was several centimeters shorter than that  $\sim$ 27 cm. In this case we split the core into intervals of 9cm, assuming that compaction happened linearly. In this way, we avoided strong artefacts related to compaction, because 0-9, 9-18 and 18-27 cm in a compacted core should come relatively close to 0-10, 10-20 and 20-30 cm in an uncompacted core. To clarify this, we added the following sentences: 'In case of soil compaction within the auger, the increment depth was adjusted linearly. For example, a compaction of three cm over the whole soil core resulted in a sampling of 0-9, 9-18 and 18-27 cm increments.

This brings me to my next point – if the warming of the soil has caused changes in the

**SOILD**
soil density particularly in the "top soil" this would cause the sampling at the warmer place to go deeper into the subsoil extracting soils that naturally (before the shift in geothermal flow) had a lower content of C. This would then be compared to the lower layer of the "none" warmed soil and we would wrongly conclude that the warming has caused loss of carbon?

Answer: Again, we agree with the reviewer that this is a problem to a certain extent: Shifts in SOC contents lead to shifts in bulk density, which should -- in an optimal worldbe considered before sampling. However, the difficulty is that such changes in bulk density are usually not known before sampling, so how can depth increments be properly adjusted? The only way is then to use a defined volume (e.g. a metal frame) and conduct already the sampling by equivalent soil mass (the approach that is mentioned below). Accounting for equivalent soil mass in the field is extremely elaborate and therefore rarely done. More importantly, it is based on fine soil (<2mm) of course, because this is where the carbon is. In this young, volcanic soil, the rock fragment fraction is extremely variable and partly very high (especially in the grassland soil >10cm). A sampling based on equivalent FINE soil mass is therefore not possible in this soil, and also bulk density values (that have been measured before) are hard to interpret. Nevertheless, the reviewer brought up an important point that is ignored in many studies measuring gradients in SOC contents (or even stocks). We have one argument that the observed loss of SOC is mostly really related to the warming effect: Along the soil profile of grassland and forest soils, the strongest gradients in SOC content with depth occur in the upper cm. The deeper in the profile, the less steep the gradient. This means, that a bulk density related sampling bias would show up most extreme in the topsoil (i.e. the relative loss of SOC should be higher in 0-10cm than in 20-30cm). However, in our case the opposite was the case: Relative losses were more pronounced in the subsoil. We now address this issue in the following sentences in the discussion: Also, those structural changes did most likely lead to a certain sampling bias and thus a slight overestimation of SOC losses: A sampling of fixed depth increments ignores the fact that depth increments change with changes in bulk density. Therefore, the

**SOILD**
depth increments sampled in the higher warming intensities do not exactly match the depth increments sampled in the lower warming intensities. However, this effect is expected to be more pronounced in the topsoil, were the SOC depth gradient is largest and thus a shift in reference soil depth would have the strongest impact on bulk SOC content. However, relative losses in SOC were even more pronounced in the subsoil, indicating that the sampling bias was might have been small. However, it should be mentioned that a mass-based instead of a depth-based sampling (Don et al. 2020) or at least an a-posteriori soil mass correction (Ellert and Bettany et al. 1993) would be indispensable to accurately estimate SOC stock changes."

Bringing me to my third point –as one of the main objectives of the study clearly is to assess losses of C and also quantify the losses, we need to be sure we that what we compare are comparable. Studies like this should be done by comparing equivalent mass (See Ellert, B. H. and J. R. Bettany (1995). "Calculation of organic matter and nutrients stored in soils under contrasting management regimes." Canadian Journal of Soil Science 75(4): 529-538 or others more recent paper). Also the 10-20 cm was not analysed, understandable from a resource point of view (many time-consuming and expensive analysis), but a simple analysis of SOC + weighing of the total dry sample would have added valuable information particularly for interpretation in a climate change context.

Answer: It is correct that for calculating SOC stocks in 0-30 cm, we would need the full picture: bulk densities and SOC contents in all three depth increments. We could then also apply the mass correction as suggested by Ellert and Bettany. However, the study was not designed to measure stocks, which has its own difficulties in this specific soil, but contents. Rock fragments were not determined, total soil mass was not measured. We also think that relative loss in SOC content comes close to a relative loss of soil mass corrected SOC stock (because the differences in bulk density and total fine soil mass will be vanished by mass correction).

The most important results in this study is the relative change in proportion of C in the

**SOILD**
different fraction with increasing temperature and this should be more in focus than the loss of C

Answer: The manuscript has three clearly formulated objectives and we think that each of these aspects is important. Also, in the discussion we argue that the change in C fraction distribution and structure change (i.e. aggregate breakdown) is directly related to C loss, it is thus hard not to focus on C loss as such, although it is only based on contents.

All analysis were done on the fraction < 2mm, but it would have been interesting to know the proportion of the coarse fraction.

Answer: This is correct and we think the same, therefore a comprehensive aggregate fractionation (including aggregates >2mm) is ongoing, at least for the grassland. Here, we used the chance to apply the very same fractionation procedure as used in an earlier publication in the grassland soil also to the forest soil. Only in this way, a direct ecosystem comparison is possible.

3) The study appears to focus on loss of C from the soil with warming, however there is little information in change in input of C to the system. Warming of the soil may have had an influence on the forest growth/productivity and litterinput. Some more information on the vegetation would have been appreciated. In this study the litter /O horizonis removed – it would have been nice to at least know how thick it was at the different locations? Several papers have been written with data from this experimental area–surly some information could have been extracted from these not just giving references and leaving the reader to find out for themselves. Additional information on mineral-ogy/alternatively selective extraction of different oxideswould also aidthe discussion on C stabilization mechanism. Also, DOC and pH normally are correlated – it would have been nice to have some pH measurements to go with the top and subsoil samples. As this warming is by geothermal heating, I am naturally curious to how this is affects the the top- and subsoil, are there any effects on soil moisture, any gradient between the

SOILD
**two layers.**

Answer: Yes, there was also a vegetation response, which could partly drive SOC losses. However, estimating C inputs to the soil is extremely complicated in this setting, because not only productivity, but also biomass turnover needs to be estimated. Those studies are underway in both ecosystems and are the focus of two PhD projects. Concerning this, we have added the following sentences, using unpublished data from the database (section 4.1): "In fact, root biomass in 0-10 cm decreased in both ecosystems (data not shown), leading to weak positive correlations ( $R^2$ =0.37 for forest and  $R^2$ =0.29 for grasslands) of SOC and root biomass. Also above ground plant litter tended to decline in both ecosystems. This suggests that SOC losses were partly driven by decreasing C input with warming and not by increased microbial activity alone. However, a clear picture on absolute C inputs in the experimental plots is not available yet, since it needs to consider NPP and biomass turnover at the same time." Much of the data that is needed to understand the whole picture better is still in the pipeline and unpublished. As for mineralogy and oxides, we must admit that those have never been measured so far. Surely this would be interesting, but in our view there will be only minor changes in mineralogy with warming, so it might not be the most important parameter to explain the responses in this study. Regarding soil pH: We agree that an increased pH could also explain that observation. We have now added the following sentence: "However, also soil pH is acknowledged to affect DOC formation (Kalbitz et al., 2000), which might be another possible explanation for the observed increase in the proportion of DOC: in both ecosystems, soil pH increased by up to 0.5 units in the highest warming intensity (Sigurdsson et al., 2016).". Soil moisture (only in 0-5 cm) has been measured in both ecosystem between April and August 2016. Interestingly, the difference between treatments was small and not related to warming. Data are presented in Sigurdsson et al. 2016 Journal of Icelandic agricultural sicences. We do not have information on a potential depth gradient.

4) The use of the term "topsoil" and "sub soil" when this refers to 0-10cm (topsoil) and
20 -30 (sub soil) is ill-conceived. I normal soil terminology both these layers refer to topsoil. Readers with an interest in C subsoil – non surface layers will perhaps be misled. Why not simply us "Upper" and "Lower" or even better were there any genetic differences A horizon – B horizon?

Answer: We agree that the term subsoil is potentially misleading when talking about soil within the top 30 cm of soil. However, the investigated soils, especially in the grassland, were very shallow and of course there are soils with shallow subsoils. We think that this is a matter of definition and define top and subsoil already in the abstract. Therefore we do not see that a change from topsoil to subsoil into "Upper soil" and "Lower soil" or "Upper" and "Lower" would improve the manuscript. Also, we did not sample along genetic horizons, especially not representatively, therefore we cannot change the fixed depth increments into pedogenetic horizons. By the way, pedogenetic horizons are also very hard to determine in these young Andosols.

5)Ecosystem comparison: I believe there should have been more focus on differences in input of C. You observe differences between the ecosystems only in the topsoil – ascribing this to the fact that the forest was planted on former unmanaged grassland. However, you also find that the forest soil has a more pronounced depletion of Cin the subsoil. Could also part of the explanation also be due to the fact that warming was geothermal – from beneath. In a situation where global warming (air warming) is the case the differences in the topsoil would be equally reflected in the subsoil.

Answer: Warming is coming from below, that is correct, and it might to some extent explain the relatively stronger response. This was already part of the discussion (section 4.1). But why or how could it explain that there is a distinct ecosystem response in the topsoil but not in the subsoil? For us, it is more likely that this is due to the fact that forest and grassland subsoils were generally very similar (in SOC and fraction distribution) and so responded also similarly to warming.

---

## Author Comment (AC3) · 13 Nov 2019

The manuscript by Poeplau et al., investigates the effects of long-term soil warming on SOC fractions. Their carbon content as well as their relative distribution in response to warming are being discussed for two different soil depths and ecosystems. As I myself am working on warming effects on soil microbial communities in connection to biogeochemical cycles, I was very pleased to read about warming effects on the abiotic components of soil. Especially long-term in-situ warming experiments are rare and extremely valuable to study the mechanisms and concepts behind various warming effects. I completely agree with the authors that strong systematic gradients in SOC

content (in the same soil) can provide an important framework to improve our understanding of SOC dynamics. I believe that this study adds some very valuable aspects the research field of soil warming. This paper and some there presented ideas could also provide the basis for hypotheses that could be targeted in other future mechanistic studies. The manuscript is well written and structured. Most parts of it are easy to follow, however I hereby want to suggest some minor and detailed revisions in order to improve the manuscript in terms of its readability and understandability.

Answer: We thank the reviewer for these very positive statements and many helpful comments that helped to improve the manuscript.

Personal comments: Out of personal interest, I would like to ask if you have a suggestion to why both ecosystem types approach to more or less the same SOC content in response to long-term warming (Fig. 4 and 5)? The levelling off of (absolute) SOC losses from different soil types (with different native C contents) to a sort of threshold level is very interesting to me.

Answer: Indeed, this is interesting, but we have to keep in mind that the soil type is not that different, it is the land cover that differs. We added the following sentences to section 4.4: 'Finally, SOC contents in both ecosystems approach a similar baseline in the highest warming intensity. This might indicate that the specific amount of biogeochemical persistent SOC does not depend on land cover or vegetation type, but is rather controlled by mineralogy.'

I also like the idea of decreasing pore space with warming (due to aggregation loss). I myself often observed a decrease in microbial biomass with warming, which I now start considering to link to the suggested decrease in pore space and microhabitats. Therefore, one can see the need of better exploring structural soil changes upon warming in order to better interpret biotic responses.

Answer: Yes, we fully agree and hope to go into more detail regarding soil structure (of undisturbed soil columns) in the future.

Out of curiosity I also want to ask if you observed any changes in the vegetation cover after several years of soil warming? And if yes, if there is data on plant communities available or a paper covering that aspect? As far as I know, changes in vegetation with warming are quite often observed and I wonder about associated impacts on SOC contents and fraction distribution. E.g. if observed effects like the loss of aggregation could be associated with a change in the vegetation structure? Out of personal interest, is the light ultrasonic treatment more effective and more representative for natural conditions than the slacking treatment?

Answer: There is one paper out on phenology in the grassland (Leblans, Niki, Bjarni D Sigurdsson, Sara Vicca, Yongshuo Fu, Josep Penuelas, Ivan Janssens (2017). Phenological responses of Icelandic subarctic grasslands to short-term and long-term natural soil warming. Global Change Biology 23(11), 4932-4945. doi: 10.1111/gcb.13749) and the consortium has gathered quite some data on vegetation dynamics and biomass, much of it is however still unpublished or not available in a form that it can be properly used, and currently there are two PhD students working on those NPP issues, aboveground and belowground in both ecosystems. However, we agree that vegetation responses should be mentioned to some extent, because they are likely to drive SOC dynamics. We now added the following sentences to section 4.1: "In fact, root biomass in 0-10 cm decreased in both ecosystems (data not shown), leading to weak positive correlations ($R^2$=0.37 for forest and $R^2$=0.29 for grasslands) of SOC and root biomass. Also aboveground plant litter tended to decline in both ecosystems. This suggests that SOC losses were partly driven by decreasing C input with warming and not by increased microbial activity alone. However, a clear picture on absolute C inputs in the experimental plots is not available yet, since it needs to consider NPP and biomass turnover at the same time." Regarding the slacking: This is a very good question and slacking is usually done in the context of aggregate size fractionation or any questions related to aggregate stability. So in our case, slacking and an aggregate size fractionation would have been interesting as well (and is currently done by other groups working on the ForHot experiment). However, this was not clear before we started the

fractionation work in the grassland and the beauty of this study is that we could repeat the exact same method for the forest.

Specific comments: Short comment to the title: I am not sure if you can really call it a "deteriorated" soil structure. I would rather use the term "changed" or "affected" because it does not imply a judgement. Also. The authors sometimes refer to boreal systems throughout their manuscript. However, I much better like the term subarctic in the context of the study site. Throughout the manuscript, I found an inconsistency in terms of SOC terminology as it is stated on page four and also within the use of associated units. In order to improve the readability and the understandability of the text, I would also welcome to be more specific and explicit when writing about SOC changes e.g. SOC contents, concentrations, fractions, mass et cetera. For detailed comments about alternative phrasing suggestions see below.

Answer: We changed the title of the manuscript. Following suggestions also of the other reviewer, we now removed soil structure from the title, mentioned the very specific soil type (Andosol) and included that two different ecosystems were evaluated: "Strong warming of a subarctic Andosol depleted soil carbon and aggregation under forest and grassland cover". Regarding the word 'boreal', we have replaced it by subarctic or deleted it where appropriate. The suggestion on being more specific about the use of SOC was also realized (see below). Page 1: Line 14: Five different SOC fractions were isolated and their re-distribution as well as the amount of stable aggregates was assessed to link SOC to soil structure changes.

Answer: We agree and used the suggested sentence in the abstract.

Line 16: Soil warming had depleted SOC concentrations in forest bulk soil by...

Answer: We agree but used contents instead of concentrations.

Line 24:...indicating an indirect protective effect of SOC on aggregates...

Answer: Here, the rationale was actually the other way around. We now slightly

changed the sentence to clarify this: '...indicating an indirect protective effect of aggregates >63 $\mu$m on SOC.'

Line 25: Topsoil changes in total SOC content and fraction distribution...

Answer: Changed accordingly.

Line 27:...in the response of subsoil SOC content and fraction distribution...

Answer: Changed accordingly.

The authors write in the abstract that no ecosystem effect was observed. However, this was confusing to me, as in my understanding Tab. 3 shows significant effects.

Answer: In the abstract, we are referring to the interactive effect of warming and ecosystem, not to the ecosystem effect (interaction effect is also given in the table). The question was, if the two ecosystems responded differently to warming. We modified the sentence to clarify that: 'However, no ecosystem effect on the warming response of subsoil SOC content and fraction distribution was observed.'

Line 32: Could you please specify if the stated temperature increase refers to air or soil temperature?

Answer: This is air temperature and we added that information.

Page3 Line 3: I believe that the statement about permafrost soils is out of place here as no permafrost is occurring at the investigated site.

Answer: We agree and deleted this part of the sentence. It now reads: 'The highest SOC stocks are located in high northern ecosystems (Tarnocai et al., 2009).'

Line 32: It was a bit hard to understand to how many samples you refer in your manuscript. Could you please state the explicit number of samples taken? Also, are all mentioned five transects situated in the investigated forest? Please also indicate the number of samples in your graphs.

Answer: The information was given above, when introducing the design of the experiment. However, it seamed a bit hidden because also the other reviewer did not see that. We therefore added that information again when describing the sampling procedure. The sentence (L30) read as follows: 'In late April 2018, i.e. almost exactly 10 years after the warming was initiated, mineral soils of all permanent forest plots (six warming intensities, five replicates each) were sampled.'

Page 4 Line 6: Distinct responses to warming were thus expected. Could you make an explicit statement what you were expecting?

Answer: Instead of going into more details here and setting up hypotheses in the middle of materials and methods, we decided to delete this small sentence. The method description reads more fluent now.

Line 15: Unit is missing for SPT (1.8 g cm3 -1)

Answer: Changed accordingly.

Line 33: Out of personal interest - what is the variability to the average mass recovery?

Answer: We added standard deviations to the given recoveries: 'Average mass recovery was 97$\pm$2%, average C recovery was 99$\pm$21%'. Of course, mass recovery was less variable.

Page 5 Line 11: Would you assume a positive or negative correlation between the poured bulk density and SOC in the SA fraction?

Answer: We changed the sentence slightly into: '...and hypothesized that _i would be negatively correlated to SOC content in the SA fraction in particular.'

Line 31: Do you think that absolute SOC losses are higher in topsoil because of a higher "native" C concentration?

Answer: This is likely because relative SOC losses were similar (or even higher in the subsoil). We however do not speculate about this here, because this is not the

discussion part.

Page 6 Line 1: Please state here that you talk about SOC contents in bulk soil.

Answer: Done.

Line 11: The depletion of SOC content lead to a changed relative distribution...

Answer: Done.

Line 12: The ANOSIM revealed... Please state here that you talk about topsoil findings.

Answer: Done: 'The ANOSIM revealed that a warming intensities of 5.8 and 17.5°C were necessary to significantly change topsoil SOC distribution (Tab. 2).'

Line 13:...fraction distribution was only significant from the unwarmed reference at a warming intensity of 5.8C. I was wondering about the + 2.7°C treatment (see Tab. 2)?

Answer: That is correct, in the subsoil significant differences were also found for 2.7°C. This was now included.

Line 15:...SOC in the POM and SA fractions, which were strongly depleted with warming (Fig 1). Please mention here, that this was the case in both depth increments.

Answer: In fact, in this specific sentence we are talking about the topsoil only. We therefore started the sentence now with 'In the topsoil...'.

Line 28:...the relative mass proportion of rSOC was expected to increase...

Answer: Done.

Line 30: Could you please give the p-value for the mentioned regression between rSOC and total SOC in the SC fraction?

Answer: Done. It was p<0.001.

Line 36: Could you please state the p-value for the significant negative relationship between the proportion of SOC in SA and the proportion of SOC in DOC

Answer: Done. It was p=0.002.

Page 7 Line 5: You mention similar SOC contents for subsoil forest and grassland soils. Is there more information about that e.g. an ANOVA?

Answer: We agree that similar is a vague term and actually the ANOVA (as presented in Tab.3 gave significant differences between ecosystems also in the subsoil. We have now rephrased the sentence to be more precise. It now reads as follows:'Also, the difference between ecosystems in subsoil SOC contents was less pronounced than in the topsoil.'

Line 14: You mention that POM in forest soils responded more negatively to warming than POM in grassland soils. Was this normalized to their respective C contents?

Answer: It is inferred from the slope of the regressions in Fig.4, which shows that forests start at a higher level than grasslands and both end up with a similar amount. So yes, it is true in absolute and relative terms.

Line 17: I would have appreciated it, if you mentioned earlier on that the warming in the grassland soils was only 6 years compared to the 10 years of warming in the forest.

Answer: This was mentioned in material and methods. Now on page 5, lines 12-14.

Line 20:...we found a strong negative correlation of bulk soil SOC content and poured BD. Please also give the R2 here.

Answer: Done.

Line 34: According to Tab. 1 the relative change in topsoil SOC content is -3.6% not -2.7% as stated.

Answer: This is correct. Numbers were mixed up here. Changed accordingly.

Line 36: Do you think that the + 5.8°C is a realistic warming intensity for soil or air temperatures?

Answer: To be clear about this difference, we modified the section, which now reads as follows: 'Considering that an air temperature increase of up to 11°C until the end of the century is within the possible range of IPCC climate change projections (IPCC 2013), we assume that a soil warming intensity of up to 5.8°C can be considered realistic. For example, Zhang et al. (2005) showed that soil temperature increase (+ 0.6°C) generally followed the air temperature increase (+ 1°C) in Canada during the 20th century. At a warming intensity of 5.8 °C, the investigated soil lost 29 % (topsoil) and 37 % (subsoil) SOC in ten years.

Page 8 Line 17: You write, that the present study did not reveal tipping points. However, if I look at e.g. Fig 1. it seems to me that +5.8âŮẹC causes some abrupt changes in SOC contents and SOC proportions in fractions? Please also specify "tipping points for SOC" here (e.g. SOC contents).

Answer: This is correct, it seems like the aggregate break-down kicks-in only after 2.7°C. However, in the mentioned section we discuss bulk SOC content only. We clarified this and also added a sentence on page 9 line 24-25: 'A tipping point for aggregate-breakdown appears to be located between the warming intensities of 2.7 and 5.8°C.'

Line 22: You write that climate change is likely to strongly affect SOC stocks of boreal forests. Generally, I strongly agree to that statement. However, would question the comparability of the relatively young investigated forest on volcanic bedrock material to the biome of naturally old grown boreal forests. Upscaling to the regional or global context might be a slight over interpretation.

Answer: We conducted a soil warming experiment in a subarctic forest and find strong losses of SOC. We do not think that it is an overinterpretation to infer that climate change is likely to affect subarctic forest SOC elsewhere or as a whole. We don't state that this will happen at the same rate as we found, so we think it is ok to leave that statement in the text.

Page 9 Line 17: In the unwarmed reference soil, it accounted for the highest proportion of soil mass and SOC content.

Answer: changed accordingly.

Line 29:...we found a very strong positive correlation of SOC mass and...

Answer: changed accordingly.

Page 10 Line 2: I did not understand the context of the sentence about carbon desorption from the mineral phase. It seemed a bit out of place to me.

Answer: We changed the sentence slightly: 'Desorption of carbon compounds from the mineral phase is likely to be fostered by increased surface area, which is the case when aggregates disintegrate.' – and hope that it is understandable now: break-down of aggregates increases the surface area that is exposed to water.

Line 8: According to your definition on page 4 the unit of SOC content should be (g C kg -1).

Answer: Done

Page 11 Line 7: You might rephrase the sentence to: Changes in SOC concentrations and the relative distribution of fraction masses in the grassland soils have been previously investigated.

Answer: Done.

Line 8 to Line 11: The fact that there is no difference in subsoil SOC dynamics... might indicate that the same mechanisms of SOC depletion were involved in both ecosystems.

Answer: Done.

Line 23: The sentence is very long, maybe you could split it apart.

Answer: We shortended the sentence into: 'Therefore it seems likely that amount and

fraction distribution of SOC drove the ecosystem specific warming response in the topsoil.'

Line 37: You might rephrase the sentence to: Differences in the relative distribution of SOC fractions and their respective SOC concentration in response to warming have only been found in the topsoils of both examined ecosystems.

Answer: We rephrased the sentence as follows: 'Differences in the warming response of bulk SOC and SOC fractions between ecosystems have only been found in the topsoil,...'

Specific comments about graphs and tables:

1) Tab. 2: In the table description you write about testing "differences in SOC fraction distribution". Do you mean the relative mass distribution of SOC fractions?

Answer: Well, it is about how SOC is distributed in different fractions (reformulated to 'testing differences in the distribution of SOC in investigated fractions'). Fraction mass is not correct here, because in this paper fractions mass refers to the total soil mass in a specific fraction, not carbon.

2) Fig.1: is missing the a)b)c)d) notation in the individual graphs. Also the unit of the x-axis of a) and b) should be changed to SOC content (g C kg-1).

Answer: Done.

3) Fig.3: For me, the graph would be easier to understand if the title of the x and y axis would be changed to "percentage of total SOC in SA" and "percentage of total SOC in DOC". Also a p-value is missing in the graph as what is the depicted error range (95% confidence interval?).

Answer: Changed accordingly.

4) Fig 4: I am a bit confused if the scatter plots show SOC masses or SOC contents of the fractions in response to warming. What is the depicted error range? Moreover,

some relationships seem rather curvilinear to me than linear.

Answer: It is the carbon content as defined in m&m section, which you might understand as carbon mass actually (g C in the fraction per kg soil). It is confusing, because you can look at the data from different angles and at another point, we are talking about SOC concentration, which refers to g C in the fraction per kg fraction. The error is the 95% confidence interval and that was now added to the caption. Also, we admit that it was a bit oversimplified to use a linear fit in all cases. We now selected the best fit for each case using AIC, deciding between linear and logarithmic fits and also adjusted the statistics section.

5) Fig 5: Please change the title of the the y axis to subsoil instead of topsoil. I moreover have the same small issues with the graphs as mentioned for Fig. 4.

Answer: Changes accordingly.

6) Fig. 6 shows regression models. Please indicate the p-values here. 6a) The unit of SOC content should be (g C kg soil -1). 6b) The unit of SOC concentration should be (g C kg fraction -1). 6c) Please change the title of the x axis to " soil mass in stable aggregates". 6d) The shown relationship looks more curvilinear than linear to me.

Answer: We have added the p-value of <0.001 in the caption once, because this applied to all four regressions. We have changed the axes accordingly and regarding 6D: In fact, the linear fit was the best. The optical impression is a bit misleading, because of three slightly outlying points in the subsoil.

Questions provided by SOIL: 1) Does the paper address relevant scientific questions within the scope of SOIL? Yes, I think so. 2) Does the paper present novel concepts, ideas, tools, or data? The paper shows new interesting data on physical soil structure changes in responses to warming. However, to my state of knowledge, no new tools were involved. The manuscript provides some new concepts and ideas e.g. the proposed mechanism of warming leading to SOC loss (via enhanced microbial activity) which then results in the loss of stable aggregates. I also very much appreciate the proposed idea that the slope of the regression line between SOC and bulk density might be a useful indicator for aggregation affinity in unmanaged soils.

3) Does the paper address soils within a multidisciplinary context? n.a.

4) Is the paper of broad international interest? The scope of physical soil fractions and their response to warming seems of broad interest. It represents a framework of many biological responses to higher temperatures.

5) Are clear objectives and/or hypotheses put forward? Three objectives are stated clearly on page three and then also addressed in the results and discussion of the paper.

6) Are the scientific methods valid and clear outlined to be reproduced? Yes. Especially the SOC fractionation protocol is described in great detail and could be repeated in ourlab too.

7) Is the soil type/classification adequately described? Yes, most of the general information on soil types is given in the text.

8) Are analyses and assumptions valid? Yes.

9) Are the presented results sufficient to support the interpretations and associated discussion? Yes.

10) Is the discussion relevant and backed up? In general yes. For detailed comments see above.

11) Are accurate conclusions reached based on the presented results and discussion? Yes.

12) Do the authors give proper credit to related and relevant work and clearly indicate their own original contribution? Yes.

13) Does the title clearly reflect the contents of the paper and is it informative? Yes. The title clearly reflects the later proposed mechanism of aggregate break-down which follows SOC loss that was caused by warming. I especially like that the title includes the term "subarctic" and not "boreal".

14) Does the abstract provide a concise and complete summary, including quantitative results? Yes.

15) Is the overall presentation well structured? Yes, I like the tripartite structure of the paper (1-warming effects on forest SOC and its fractions, 2-forest vs. soil SOC in response to warming, 3- soil structural changes). The focus on those three topics can be found in the introduction, results and discussion part.

16) Is the paper written concisely and to the point? To my understanding the manuscript is mostly concise. However, sometimes the sentences were hard to follow (too long) and not precise enough to understand what the authors meant. This holds especially for "SOC-terminology and SOC units" – see detailed comments above).

17) Is the language fluent, precise, and grammatically correct? Mostly yes. However, some sentences are relatively long and thus hard to follow. This is especially the case in the discussion part.

18) Are the figures and tables useful and all necessary? The figures are nice and useful. For detailed suggestions see comments above.

19) Are mathematical formulae, symbols, abbreviations, and units correctly defined and used according to the author guidelines? Yes. In the context of units, please see detailed comments above.

20) Should any parts of the paper (text, formulae, figures, tables) be clarified, reduced, combined, or eliminated? I think that the sampling procedure (amount of taken samples and analyzed) could be described in more detail.

21) Are the number and quality of references appropriate? Yes.

22) Is the amount and quality of supplementary material appropriate and of added value? Yes

---

## Author Response (AR2)

Dear Editor,

thank you for your positive decision from today. We are happy that this manuscript approaches final publication, and respond to your suggestions as follows:

Abstract
- 'However, specific ecosystem responses to warming are understudied'. maybe better: specific soil responses to warming under different ecosystems are understudied

Answer: Changed accordingly.

- 'We used a natural geothermal soil warming gradient in an Icelandic spruce forest (0-17.5 °C warming intensity)' should be: We used a natural geothermal soil warming greadient (0-17.5 °C warming intensity) in an Icelandic spruce forest on Andosol

Answer: Changed accordingly.

Conclusion:
- 'Using a strong geothermal warming gradient, we highlighted the critical role of SOC for soil structure.' Should be: Using a strong geothermal warming gradient we investigated soil structural changes and their effect on different SOC pools.

Answer: We have reformulated this sentence in a rather different way and hope that you can agree to that as well: "…we found a clear link between SOC losses and soil structural changes".

-'changes in the soil abiotic environment '- may be better: changes in the microbial habitat and possibly abiotic soil properties

Answer: Changed accordingly.

Sincerely Yours,

Christopher